# When Attention Collapses: How Degenerate Layers in LLMs Enable Smaller, Stronger Models

**Sunny Sanyal**[*]
*The University of Texas at Austin*

**Ravid Shwartz-Ziv**
*New York University*

**Alexandros G. Dimakis**
*UC Berkeley*

**Sujay Sanghavi**
*The University of Texas at Austin*

**Reviewed on OpenReview:** *https://openreview.net/forum?id=2zQnObUoPf*

## Abstract

Large Language Models (LLMs) are known for their performance, but we uncover a significant structural inefficiency: a phenomenon we term attention collapse. In many pre-trained decoder-style LLMs, the attention matrices in deeper layers degenerate, collapsing to near rank-one structures. These underutilized layers, which we call *lazy layers*, are redundant and impair model efficiency. To address this, we introduce Inheritune, a simple yet powerful training recipe designed to build smaller, stronger language models. Inheritune initializes a compact model by inheriting the potent early layers from a larger pre-trained model and then progressively trains and expands it. Our experiments on various models, including the GPT-2 family, demonstrate that models trained with Inheritune can match or even surpass the performance of their larger counterparts, despite having significantly fewer layers. This work presents a novel path toward model compression by design, enabling the creation of compact, yet highly performant language models. Code is available at https://github.com/sanyalsunny111/LLM-Inheritune.

## 1 Introduction

Large Language Models (LLMs) are composed of stacks of decoder-style transformer blocks (Vaswani et al., 2017). As the model grows in size, the model capacity and performance typically improve (Kaplan et al., 2020; Hoffmann et al., 2022). A substantial fraction of the total parameters is devoted towards adding more transformer blocks to increase the depth. Each block or layer in the stack refines the representations learned by the previous blocks, allowing the model to develop a nuanced understanding of the input data.

A transformer block primarily consists of a self-attention module and a feed-forward network (FFN, also referred to as an MLP). Among these, the causal self-attention mechanism (hereafter referred to as attention) is arguably the most critical component. It enables the model to combine token embeddings as a weighted linear sum of attention scores, effectively capturing long-range dependencies and contextual relationships within text. However, as models become deeper, they often exhibit a phenomenon known as attention degeneration, characterized by a collapse in the rank of the attention matrices. While prior studies have analyzed rank collapse in simplified transformer settings (Dong et al., 2021; Noci et al., 2022; He et al., 2023), this phenomenon has not, to our knowledge, been systematically explored in standard decoder-only LLMs. A

---

[*]Corresponding author: sanyal.sunny@utexas.edu

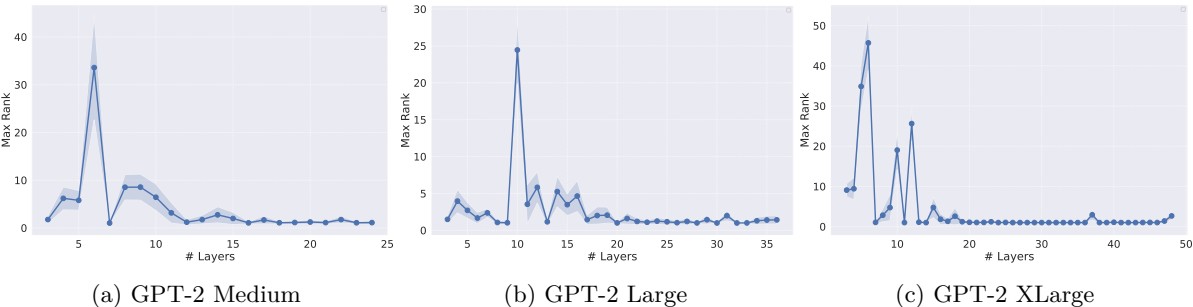

|  |  |  |
|---|---|---|
| (a) GPT-2 Medium | (b) GPT-2 Large | (c) GPT-2 XLarge |

Figure 1: **In decoder-style LLMs, attention matrices in deeper layers often degenerate to near rank-1, limiting their ability to learn meaningful representations.** We compute $\text{MaxRank}^{(l)}$ (averaged over $N = 100$ randomly selected sequences each with $T = 100$ tokens) for each layer $l$ using the OpenWebText validation set. Our rank analysis of 24-layer GPT-2 Medium, 36-layer GPT-2 Large, and 48-layer GPT-2 XLarge models reveals that attention matrices in many deeper layers collapse to near rank-1.

formal discussion of attention degeneration is provided in Section 2. In this paper, we conduct a detailed empirical analysis of attention degeneration in the GPT-2 family of LLMs (Radford et al., 2019), including GPT-2 Medium (355M), GPT-2 Large (770M), and GPT-2 XLarge (1.5B). Our analysis reveals that many deeper layers in these models exhibit predominantly rank-1 attention matrices across most attention heads within a layer. This suggests that the attention mechanism loses its discriminative ability among tokens and instead performs near-uniform averaging across the sequence. We refer to layers in which all attention matrices degenerate to near rank-1 as *lazy layers* (a more formal definition is provided in Definition G.1). In the supplementary material, we further extend this analysis to billion-sized LLMs, including LLaMA-3 8B (refer Figure 11), Falcon-7B (refer Figure 21), OLMo-1B, Cerebras GPT 2.7B and LLaMA-3 3B (refer Figure 22), to highlight that attention collapse persists even in several modern architectures.

Motivated by the new finding through our novel analysis we aim to develop performant small base language models (LMs) utilizing weights from inefficient larger base LMs without losing pre-train performance (measured by train/validation loss). A base LM is a decoder-style model trained solely for next-token prediction without additional enhancements like instruction tuning or reinforcement learning with human feedback (RLHF). Our proposal is straightforward, we start by initializing our smaller LM (target) using the first few blocks from a large pre-trained LM (reference). We then train the target model for a specified number of steps. After this initial training, we incrementally grow the target model by adding more blocks, continuing the training process until it matches or surpasses the pre-train validation loss (also val loss) of the reference model. During the growth phase, the newly added blocks can be initialized with *lazy layers* of the reference LM. We refer to this simple yet effective training approach as Inheritune.

In summary, our key contributions are as follows:

1. **Novel analysis of attention degeneration in standard decoder LLMs.** We empirically investigate attention degeneration in standard decoder style LLMs. Our analysis shows that rank-collapse in attention matrices, revealing a significant structural inefficiency in the attention mechanism of standard LLMs in deeper layers (refer Figure 1). This degeneration gives rise to what we refer to as *lazy layers*.

2. **Introduction of our training recipe Inheritune.** Building on our analysis we observe that deep LMs often fail to fully utilize their effective depth. To address this inefficiency, we propose Inheritune— a simple yet effective training recipe for developing smaller LMs without losing pre-training performance. This method involves inheriting a few early blocks from a much larger reference model and progressively growing and training the smaller model. We validate the effectiveness of Inheritune through extensive experiments on GPT-2 XLarge (1.5B), GPT-2 Large (770M), and GPT-2 Medium (355M) models, trained primarily on the OpenWebText dataset and additionally on FineWeb data.

3. **Evaluation against multiple baselines.** Models trained using Inheritune consistently outperform a wide range of baselines, including much larger models trained from scratch (see Figure 5 and Figure 7), as well as same sized models trained from scratch for twice as many steps (extended training; see Figure 6). We further compare against warm-started baselines (models initialized with pre-trained weights rather than random initialization (Ash & Adams, 2020)) as shown in Table 2.

## 2 Attention Collapse and the Emergence of Lazy Layers in LLMs

**Preliminaries:** A vanilla transformer-based model consists of $L$ transformer blocks (layers). The model operates on an input sequence $X \in \mathbb{R}^{T \times e}$, where $T$ denotes the sequence length (number of tokens), and $e$ represents the embedding dimension or model hidden size. The output of each layer $l$ is denoted as $X^{(l)} \in \mathbb{R}^{T \times e}$. Each transformer block primarily consists of two modules: a self-attention block and a feed-forward network (FFN). The self-attention mechanism enables the model to weight the relevance of different tokens in the sequence relative to each other. Specifically, for a single attention head, the attention computation is defined as equation 1.

$$\text{Attention}(Q, K, V) = \underbrace{\text{softmax}\left(\frac{QK^\top}{\sqrt{d_k}}\right)}_{\textbf{Attention matrix: } A(X)} V \tag{1}$$

where the queries $Q = XW_Q$, keys $K = XW_K$, and values $V = XW_V$ are linear transformations of the input $X$. Here, $W_Q, W_K \in \mathbb{R}^{d \times d_k}$ and $W_V \in \mathbb{R}^{d \times d_v}$ are the weight matrices for the queries, keys, and values, respectively. Typically, $d_k = d_v = \frac{d}{h}$, where $h$ is the number of attention heads. In this single-head scenario, we set $d_k = d_v = d$.

The **attention matrix** $A(X) \in \mathbb{R}^{T \times T}$ captures the pairwise attention scores between all token positions in the sequence. The softmax is applied row-wise. The attention matrix $A(X)$ is then used to compute a weighted sum of the value vectors. **Attention rank collapse** refers to the phenomenon where the attention matrices $A(X)$ of individual heads in many layers of transformer-based language models lose their expressive capacity, converging towards lower effective rank structures. Specifically, the effective rank of attention matrices significantly reduces, often approaching rank-1, limiting the model's ability to meaningfully differentiate between token interactions across positions in the sequence. Previous research by Dong et al. (2021) and He et al. (2023) has shown that in self-attention networks (SANs) without residual connections and feed-forward networks (FFNs), the rank of an attention matrix converges to rank-1 doubly exponentially with respect to the depth of the model. This phenomenon, known as rank collapse of attention matrices, results in a loss of expressive power as the attention mechanism attends to all tokens uniformly. Noci et al. (2022) showed that even with residual connections (without layernorm) attention matrices can still lose rank in deeper layers if the residual connections are not scaled by $1/\sqrt{L}$. Interestingly they also linked the rank collapse to vanishing gradients of the keys and queries in deeper layers which affects the overall trainability of the transformer based models. However, these findings do not directly apply to the standard LLMs, as transformer blocks in these models include residual connections, layernorms and FFNs, which are expected to mitigate both rank collapse and the vanishing gradient problem.

**Approximate Rank Computation of Attention Matrices** To assess the presence and severity of rank collapse within standard decoder style transformer architectures (e.g., GPT-2, LLaMA etc.), we utilize singular value decomposition (SVD) for each attention matrix $A(X) = U\Sigma V^\top$, where $\Sigma$ is a diagonal matrix containing singular values $\sigma_1 \geq \sigma_2 \geq \cdots \geq \sigma_T \geq 0$. The approximate rank (referred to as rank hereafter) of an attention matrix, parameterized by a variance threshold $\tau$, is formally computed as:

$$k^* = \min\left\{ k \in \{1, 2, \ldots, T\} \mid \frac{\sum_{i=1}^{k} \sigma_i^2}{\sum_{j=1}^{T} \sigma_j^2} \geq \tau \right\},$$

where $\tau \in (0, 1)$ represents the proportion of variance that must be captured by the top $k$ singular values. A lower value of $k^*$ indicates stronger rank collapse. In this work, we set $\tau = 0.90$.

In Figure 1, we present the layer-wise analysis of rank of GPT-2 models. For this analysis, we computed $A(X)$ using $N = 100$ sequences selected at random from the validation set of OpenWebText with 4M tokens, each with a sequence length of $T = 100$ tokens across all attention heads within each layer. We then define the average approximate rank for each head and layer as $\text{Rank}^{(h,l)} = \frac{1}{N} \sum_{n=1}^{N} k_{n,h,l}^*$. Subsequently, we aggregate this metric per layer by taking the maximum rank across heads: $\text{MaxRank}^{(l)} = \max_h \{\text{Rank}^{(h,l)}\}$. As demonstrated in Figure 1, $\text{MaxRank}^{(l)}$ reveals that many deeper layers exhibit attention matrices that are predominantly near rank-1. We highlight that this rank collapse occurs in GPT-2 Medium, Large, and XLarge models, which are widely used modern LLMs thereby extending the limited findings of Dong et al. (2021) and Noci et al. (2022). We further visualize attention matrices from representative potent and *lazy layers* of GPT-2 XLarge in Figure 12. Overall, the degeneration of attention matrices in deeper layers provides quantitative evidence for the existence of *lazy layers*. Specifically, we observe that some deeper layers exhibit a near-complete rank collapse of attention matrices across all heads, suggesting potentially reduced representational capacity and less effective token mixing in these layers.

We provide an extended discussion of attention collapse in Appendix B. We analyze five modern LLMs for attention collapse (also refer Appendix G). We further assess the robustness of our findings by conducting collapse analyses on different datasets and by using an alternative metric, namely the mass of attention matrices. Finally, we perform an ablation over a range of values of $\tau$ (refer Appendix 4.3) to study the sensitivity of the rank-based analysis with respect to $\tau$.

## 2.1 The Functional Ineffectiveness of Lazy Layers

Having identified *lazy layers*, we investigate their practical utility: *Do these structurally degenerated layers retain transferable knowledge, or are they functionally impaired?* Our experiments suggest the latter.

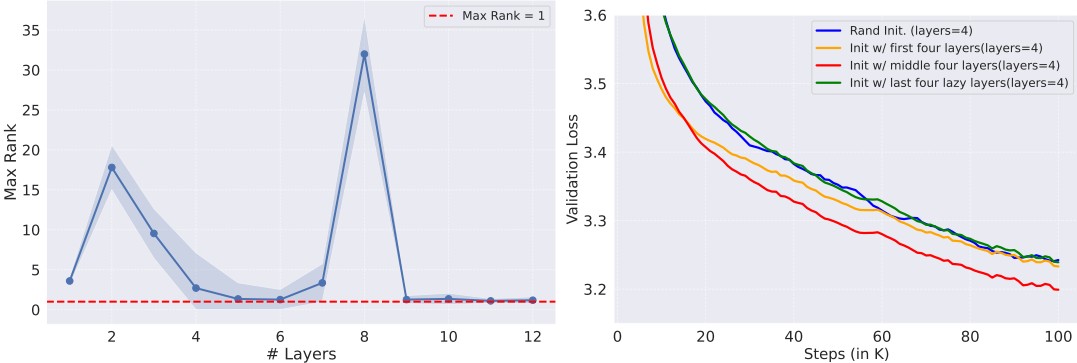

(a) Rank analysis of GPT-2 Small (12 layer).  (b) Performance of 4 layer GPT-2 small variants.

Figure 2: **Higher-rank (potent) layers transfer better.** (Left, a) Layer-wise $\text{MaxRank}^{(l)}$ of a pre-trained 12L GPT-2 Small. (Right, b) Validation loss of 4 layer variants initialized with potent layers (AvgRank $\approx 8.4 - 9.5$) vs. *lazy layers* (AvgRank $\approx 1.2$) and random weights, after training for 100K steps. Models initialized with *lazy layers* mirrors the model with random initialization. Training curves are smoothed for visual clarity.

In the first set of experiments, we trained a vanilla GPT-2 small (125M) model with 12 layers for 100K steps on the OpenWebText dataset. We then performed the rank analysis described earlier, with results presented in Figure 2. Specifically, we aggregated the approximate ranks over groups of contiguous layers using $\text{AvgRank} = \frac{1}{L} \sum_{l=1}^{L} \text{MaxRank}^{(l)}$, where $L$ is the number of layers in each group. Subsequently, we trained three GPT-2 small variants[1] for 100K steps, each initialized with a different contiguous block of four layers from the trained vanilla GPT-2 small model: (a) layers 1–4, with AvgRank = 8.40; (b) layers 5–8, with AvgRank = 9.48; and (c) layers 9–12, with AvgRank = 1.22. The last model is initialized with *lazy layers*. For comparison, we also trained another GPT-2 small variant with random initialization for 100K steps. All

---

[1]A variant shares the same configurations as the reference model but has fewer layers.

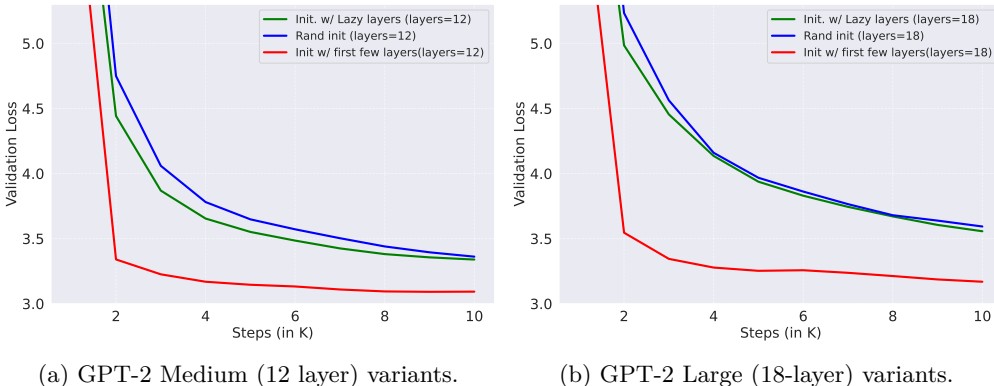

(a) GPT-2 Medium (12 layer) variants.    (b) GPT-2 Large (18-layer) variants.

Figure 3: Initializing 12-layer and 18-layer variants of GPT-2 Medium and GPT-2 Large with deeper *lazy layers* exhibiting degenerated attention results in performance comparable to random initialization. In contrast, initializing with early (high-rank) potent layers leads to substantially better convergence and generalization. Training curves are smoothed for visual clarity.

models were trained on the OpenWebText dataset. As shown in Figure 2, the model initialized with layers from the vanilla GPT-2 small model having higher AvgRank demonstrated the best performance (i.e., lowest final validation loss). Additionally, we observed that the model initialized with *lazy layers* performed very similarly to the model with random initialization suggesting that *lazy layers* contain minimal transferable knowledge. The results are also summarized in Table 4.

For the second set of experiments we utilized larger models namely GPT-2 Medium and GPT-2 Large both similarly trained for 100K steps using OpenWebText. Here we initialized a 12-layer GPT-2 Medium variant and an 18-layer variant of GPT-2 Large using *lazy layers* extracted from pre-trained 24-layer GPT-2 Medium and 36-layer GPT-2 Large models. We then trained these GPT-2 variants on the same dataset for 10K steps. For comparison, we conducted two baseline experiments where the GPT-2 variants were initialized either with the first half of transformer layers (potent layers with high AvgRank) and with random initialization. As shown in Figure 3, models initialized with *lazy layers* demonstrate poor transferability, performing similarly to models with random initialization. This provides additional evidence that *lazy layers* with fully degenerated attention, fails to learn meaningful representations.

**Theoretical Analysis.** Finally, we analyze the implications of attention rank collapse on model training from a theoretical perspective (refer Section H). Our key insight is that rank-collapsed attention head(s) impede learning by inducing vanishing gradients, effectively suppressing updates to the associated $W_Q$ and $W_K$.

## 3   Inheritune: Our Proposed Training Recipe

This section provides a detailed description of our method, key implementation considerations, and how it addresses the inefficiencies present in current architectures.

As previously established, we have identified the problem of attention degeneration and its connection to *lazy layers*, highlighting specific inefficiencies in pre-trained LLMs. In this work, we transform this challenge into an opportunity to create smaller base language models, which we refer to as target models $\mathcal{M}_{\text{tgt}}$, that achieve comparable performance with similar or lower validation loss compared to their larger, less efficient counterparts, which we term reference models $\mathcal{M}_{\text{ref}}$.

Our proposed solution builds on two key insights: (1) the early layers of deep LLMs contain a higher concentration of potent layers with high AvgRank values, making them suitable for effective model initialization, and (2) *lazy layers* can be identified, removed, or utilized in smaller numbers, then subsequently re-trained to improve overall model capacity.

---

**Algorithm 1** Inheritune: Training Recipe for Small Language Models

---

**Require:** Reference model $\mathcal{M}_{\text{ref}}$ with $L$ layers, datasets $\mathcal{D}_{\text{train}}$ and $\mathcal{D}_{\text{val}}$, steps $\mathsf{T}$
1: Copy embedding layer and LM head from $\mathcal{M}_{\text{ref}}$ to $\mathcal{M}_{\text{tgt}}$
2: Select $l$ early contiguous layers from $\mathcal{M}_{\text{ref}}$ with high AvgRank
3: Initialize $\mathcal{M}_{\text{tgt}}$ with selected layers between embeddings and LM head
4: Train $\mathcal{M}_{\text{tgt}}$ on $\mathcal{D}_{\text{train}}$ for $\mathsf{T}$ steps
5: **while** $\mathcal{M}_{\text{tgt}}$ performance $< \mathcal{M}_{\text{ref}}$ performance on $\mathcal{D}_{\text{val}}$ **do**
6:     Grow $\mathcal{M}_{\text{tgt}}$ by inheriting additional layers
7:     Train $\mathcal{M}_{\text{tgt}}$ for $\mathsf{T}$ steps
8: **end while**
9: **return** Optimized model $\mathcal{M}_{\text{tgt}}$

---

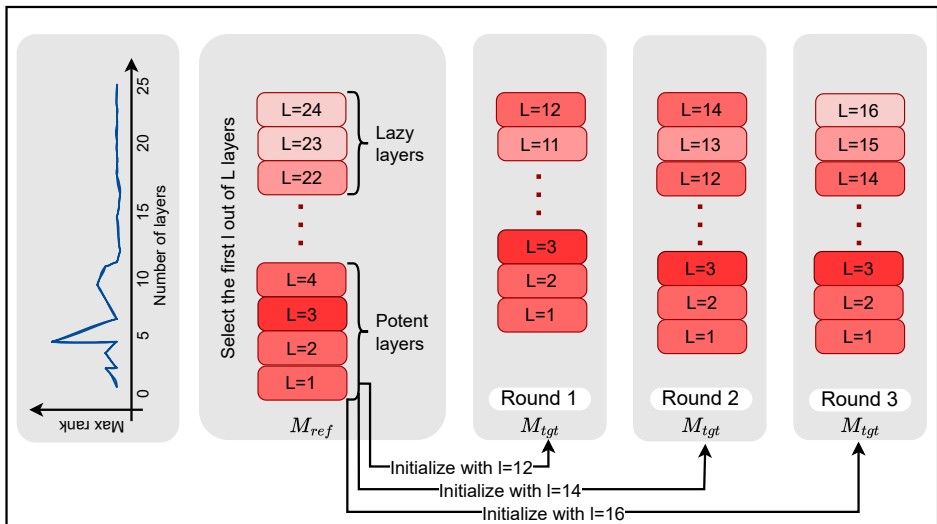

Figure 4: **Overview of the Inheritune training recipe using a 24-Layer GPT-2 Medium model example.** A smaller target model is initialized using early layers from a larger, pre-trained reference model. The target model goes multiple rounds of training while inheriting contiguous layers until it matches/outperforms the reference model. The intensity of the red color in layers correlates with $\text{MaxRank}^{(l)}$.

**Setup:** We split the dataset into a training set $\mathcal{D}_{\text{train}}$ and a validation subset $\mathcal{D}_{\text{val}}$. Next, we assume that there exists a pre-trained reference model $\mathcal{M}_{\text{ref}}$, comprising $L$ layers, represented by $\mathsf{W}_{\text{ref}} = \{\mathsf{W}_0, \mathsf{W}_1, \ldots, \mathsf{W}_{L-1}\}$ trained with $\mathcal{D}_{\text{train}}$ for $\mathsf{T}$ steps. We want to train a smaller model $\mathcal{M}_{\text{tgt}}$ with the same or better validation loss (lower is better) compared to its larger counterpart $\mathcal{M}_{\text{ref}}$.

We now present Inheritune, our proposed training recipe for efficiently developing small base language models (LMs). Inheritune operates on the principle of zero-shot initialization and progressive growth. The Inheritune process consists of three main steps, which we present below and formalize in Algorithm 1:

1. **Inherit:** Initialize $\mathcal{M}_{\text{tgt}}$ with the first $l$ out of $L$ layers of $\mathcal{M}_{\text{ref}}$, including prediction head, and token embeddings.

2. **Train:** Train $\mathcal{M}_{\text{tgt}}$ for $\mathsf{T}$ steps on $\mathcal{D}_{\text{train}}$ and evaluate on $\mathcal{D}_{\text{val}}$.

3. **Grow:** If needed, increase $\mathcal{M}_{\text{tgt}}$'s size by adding the next few contiguous layers and repeat steps 1-2 until desired performance is achieved.

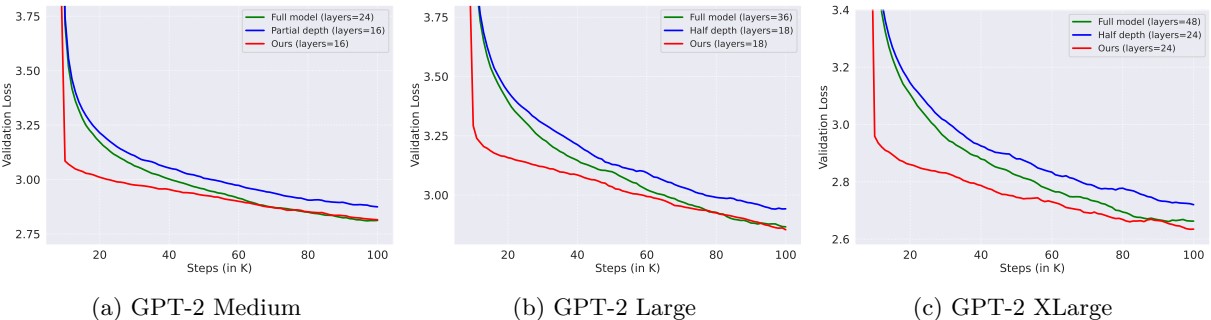

(a) GPT-2 Medium     (b) GPT-2 Large     (c) GPT-2 XLarge

Figure 5: **Models derived using Inheritune converge faster and match the final validation loss of the full-sized model, despite having much fewer layers.** Comparison of Inheritune trained models (24-layer GPT-2 XLarge variant, 18-layer GPT-2 Large variant, 16-layer GPT-2 Medium variant) against their full-sized counterparts and same sized variants trained from scratch. All models are trained for 100K steps using OpenWebText data.

With our method now formally described, we turn to empirical validation. In the following sections, we present comprehensive results demonstrating Inheritune's effectiveness across various scenarios, including different model sizes and data regimes. In addition, we conducted an in-depth ablation study to analyze the impact of initialization on performance, providing insights into the adaptability of our approach.

## 4 Experiments

We evaluate Inheritune through a comprehensive set of experiments using several GPT-2 models: a 48-layer GPT-2 XLarge (1.5B), a 36-layer GPT-2 Large (770M), a 32-layer GPT-2 Large$^{\dagger}$ (668M), and a 24-layer GPT-2 Medium (355M) (Radford et al., 2019). Table 6 provides detailed specifications of all model configurations used in our experiments.

We use two training datasets: OpenWebText Gokaslan & Cohen (2019) with 10B tokens and FineWeb (education subset) (Penedo et al., 2024) also with 10B tokens. Our experimental setup closely follows prior work (Liu et al., 2023; Sanyal et al., 2024). For models trained on OpenWebText, we report validation loss (log perplexity), while for models trained on FineWeb, we report training loss (also log perplexity).

Additionally, for models trained on FineWeb, we conduct zero-shot downstream evaluations using the `lm-evaluation-harness` (Gao et al., 2024) across five standard benchmarks: ARC-Easy (ARCE; Clark et al., 2018), LAMBADA (Paperno et al., 2016), SciQ (Welbl et al., 2017), HellaSwag (Zellers et al., 2019), and PIQA (Bisk et al., 2020). Finally, we perform a detailed ablation study by initializing each submodule within a transformer block in isolation and training it for 100K steps to identify which component contributes most to performance. All training curves are smoothed for visual clarity.

We provide experimental details of our proposed training recipe Inheritune using a GPT-2 Medium model as an example; similar procedure was applied to train other models. A visualization of the training recipe is presented in Figure 4. Our recipe for applying Inheritune involves the following steps.

1. **Reference Model:** We train a vanilla 24-layer GPT-2 Medium model (reference model) on $\mathcal{D}_{\text{train}}$ for 100K steps and evaluate its validation loss ( log perplexity) on $\mathcal{D}_{\text{val}}$. This establishes our benchmark validation loss.

2. **Model initialization:** We initialize an 12-layer model ($l = L/2$) using the reference model.

3. **Training and Evaluation:** We train the 12-layer model on $\mathcal{D}_{\text{train}}$ for $\mathsf{T}$=100K steps and evaluate its validation loss.

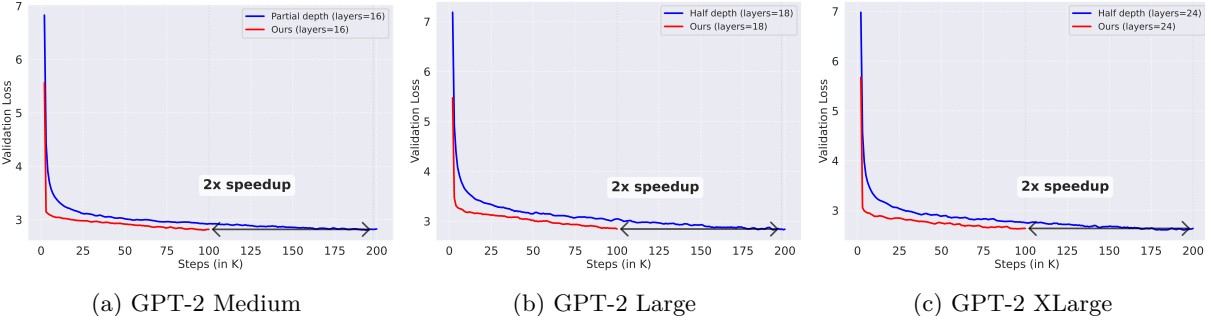

Figure 6: **Models trained with Inheritune match the validation loss of same sized models trained from scratch for twice as many steps.** We compare Inheritune-trained models (24-layer GPT-2 XLarge, 18-layer GPT-2 Large, and 16-layer GPT-2 Medium) against their same-sized counterparts trained from scratch for twice the number of steps. Inheritune models are trained for 100K steps, while the baseline models (trained from scratch) are trained for 200K steps, on OpenWebText data.

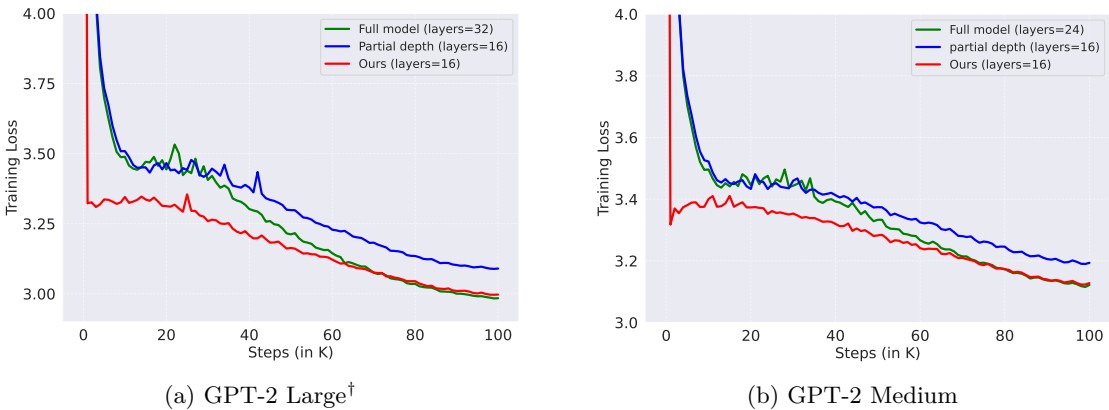

Figure 7: **Models derived using Inheritune without data repetition converge faster and match the final validation loss of the full-sized model despite using lesser layers.** Additionally, the model trained using Inheritune demonstrates data efficiency, achieving a lower validation loss in fewer steps compared to its full-sized and half-sized counterparts.

4. **Iterative Refinement:** If the smaller model's performance is inferior to the reference model, then we incrementally increase its size by adding additional layers and repeat steps 2-3 until we achieve parity with the reference model's validation loss.

We choose $l = L/2$ as the starting point and increase the model size by two layers in each round across all our experiments, aiming to minimize the number of training rounds. In principle, Inheritune should generalize to other hyperparameter choices as well.

**Baseline-I.** We compare GPT-2 model variants (i.e., models with fewer layers than their vanilla configurations) trained using Inheritune against the following baselines:

1. **Larger reference models** with more layers, trained from scratch (random initialization).

2. **Same-sized models** with the same number of layers, trained from scratch.

3. **Extended training baselines**, same-sized models trained from scratch for twice as many steps compared to models trained with Inheritune.

| Models | Recipe | Layers | ARCE | PIQA | SciQ | Hellaswag | Lambada | **Average** |
|---|---|---|---|---|---|---|---|---|
| GPT-2 Medium | rand init | 24 | 51.05 | 61.81 | 74.8 | 30.79 | 20.28 | 47.74 |
| | rand init | 16 | 49.92 | 61.92 | 73.3 | 29.56 | 19.54 | 46.84 |
| | **Ours** | 16 | 51.26 | 61.81 | 73.8 | 30.55 | 23 | **48.08** |
| GPT-2 Large[†] | rand init | 32 | 52.48 | 64.58 | 75.3 | 32.65 | 22.2 | 49.44 |
| | rand init | 16 | 50.34 | 63.11 | 75 | 30.86 | 21.56 | 48.17 |
| | **Ours** | 16 | 52.9 | 63.55 | 76.1 | 32.14 | 24.06 | **49.75** |

Table 1: **Models trained with Inheritune achieve comparable average zero-shot downstream performance to their larger reference models and surpass same-sized counterparts trained from scratch.** Downstream evaluations are performed on models pre-trained with the FineWeb dataset (see Figure 7). Performance is measured using accuracy (acc) and normalized accuracy (acc-norm) metrics, following the Open LLM Leaderboard protocol Beeching et al. (2023). We have highlighted the best average scores in **bold**.

| Models | Layers | Recipe | Steps | Pre-train Val loss ($\downarrow$) |
|---|---|---|---|---|
| GPT-2 Medium | 24 | Half-width | 100K | 3.04 |
| | 16 | Stacking | 100K | 2.84 |
| | 16 | Hybrid-stacking | 100K | 2.83 |
| | 16 | **Ours** | 100K | **2.81** |
| GPT-2 Large | 36 | Half-width | 100K | 3.06 |
| | 18 | Stacking | 100K | 2.87 |
| | 18 | Hybrid-stacking | 100K | 2.89 |
| | 18 | **Ours** | 100K | **2.80** |
| GPT-2 XLarge | 48 | Half-width | 100K | 2.77 |
| | 24 | Stacking | 100K | 2.65 |
| | 24 | Hybrid-stacking | 100K | **2.64** |
| | 24 | **Ours** | 100K | **2.64** |

Table 2: **Inheritune outperforms warm-started baselines (Baseline-II).** Comparison of pre-training validation loss for GPT-2 XLarge, GPT-2 Large, and GPT-2 Medium variants trained with Stacking, Hybrid Stacking, Half-Width, and Inheritune recipes. All baselines are warm-started, i.e., initialized with pre-trained weights rather than random initialization. Across model scales, Inheritune consistently achieves lower validation loss than the majority of warm-started baselines. The lowest validation loss (lower is better) is highlighted in **bold**.

**Baseline-II.**  Additionally, we compare Inheritune against **warm-started** baselines (Ash & Adams, 2020). Warm starting in neural network training refers to initializing a model's parameters with weights from a previously trained model, rather than starting from random initialization (a "cold start"). This setup enables a fair comparison with our approach, which also leverages prior learned representations. The warm-started baselines include stacking (Gong et al., 2019; J. Reddi et al., 2023), hybrid stacking (which initializes model layers directly from a larger reference model), and half-width (which retains all layers but reduces both the hidden dimension and number of attention heads by half, using weights from the reference model). Finally, we briefly compare Inheritune with *knowledge distillation* (Hinton et al., 2015); results are provided in the supplementary material (see Figure 17). Detailed descriptions of all baselines are presented in Section D of the supplementary material.

### 4.1 Results and Discussions

**Models trained with Inheritune outperform both larger and same sized models trained from scratch.** We present our main results in Figure 5. The 24-layer, 18-layer, and 16-layer variants derived using Inheritune from the vanilla 48-layer GPT-2 XLarge, 36-layer GPT-2 Large, and 24-layer GPT-2 Medium, respectively, achieve comparable or lower validation losses than both their full-sized counterparts and same-sized models trained from scratch, when trained for the same number of steps (100K). Our GPT-2 XLarge and GPT-2 Large variants require a single round of Inheritune training, while the GPT-2 Medium variant undergoes three rounds with 12-, 14-, and 16-layer configurations. Furthermore, as shown in Figure 6, models trained with Inheritune reach the same validation loss as same-sized models trained from scratch in approximately half the number of training steps. Moreover, for the GPT-2 Medium and Large variants, Inheritune achieves a strictly lower loss floor that same-sized models fail to reach even when trained for twice as many steps. A tabular summary of these results is provided in Appendix Table 5.

We conducted additional training experiments, using a high-quality training data namely Fineweb. We trained a custom 32-layer GPT-2 Large† (668M) and a 24-layer GPT-2 Medium (355M) reference model from scratch. Next, we trained two 16-layer variants: one derived from GPT-2 Large† and the other from GPT-2 Medium, using their respective reference models following Inheritune. For comparison, we also trained 16-layer baseline models from scratch. All models were trained for 100K steps, and training loss was used to evaluate pre-training performance. We observe thematically consistent results: as shown in Figure 7, the 16-layer variants trained with Inheritune consistently match the performance of their full-sized counterparts and outperform same-sized baselines, both in terms of training loss and zero-shot downstream evaluation. Downstream results are provided in Table 1. Model configurations and training hyper-parameters are detailed in the supplementary material (refer Section F).

**Models trained with Inheritune outperform all warm started baselines.** In Table 2, we compare GPT-2 XLarge, GPT-2 Large, and GPT-2 Medium variants trained with Inheritune against same-sized variants trained with stacking, hybrid stacking, and half-width baselines. The half-width baseline performs poorly, revealing the limitations of naive width reduction. While stacking and hybrid stacking demonstrate reasonable performance, they still fall short compared to Inheritune. Across all cases, Inheritune consistently outperforms these baselines, highlighting its effectiveness as an initialization strategy, with a single exception in the GPT-2 XLarge case where it matches one baseline. For a detailed view of the training curves across all methods, refer to the training curves in supplementary Figure 18.

### 4.2 Inheritune Mitigates Attention Collapse

We attribute the success of Inheritune, to its ability to mitigate attention collapse, thereby leading to fewer *lazy layers* after training. In Figures 8 we juxtapose the attention rank patterns of the vanilla and Inheritune-trained GPT-2 Medium. Notably, none of the GPT-2 Medium variants exhibit *lazy layers*. A similar analysis is conducted with GPT-2 XLarge (refer supplementary Figure 13).

The corresponding attention patterns for GPT-2 Medium, shown in Figure 9, further corroborate our observation. The attention patterns for both a vanilla 24-layer model trained from scratch and a 16-layer model trained using our proposed method, Inheritune. Note just for the sake of better visualization we visualized full attention and not causal attention, in practice GPT-2 models compute causal attention. We computed these attention matrices using randomly selected strings from the validation set of OpenWebText and took 40 tokens averaged over 3 runs. In the 24-layer model trained from scratch (top row of Figure 9), we observe a clear progression in attention patterns. The early layers (L4 and L7) exhibit dense structured attention patterns. In contrast, the deeper layers (L20 and L22) display more uniform patterns, indicating a loss of focus (attention). This uniformity is a hallmark of *lazy layers*, where the attention mechanism loses its ability to selectively focus on specific relevant tokens. In contrast, our 16-layer model trained with Inheritune (bottom row) demonstrates more focused and effective attention patterns, even in its later layers (L11 and L15). This striking difference suggests that our method makes the model more attentive and addresses attention collapse, potentially leading to more efficient models in compact size.

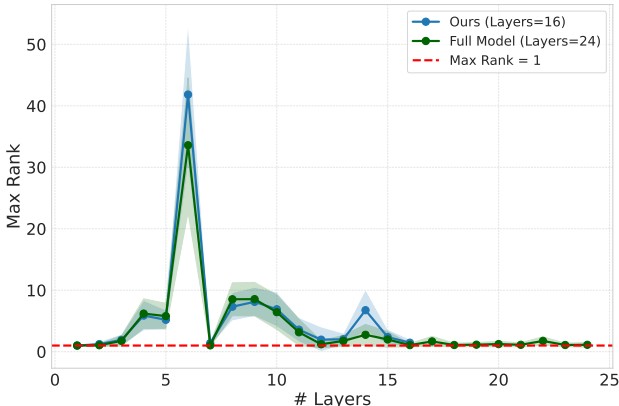

Figure 8: **Rank collapse in deeper layers and its mitigation through Inheritune.** The maximum (max) rank across all attention heads for each layer is plotted, following the methodology in Figure 1. Analysis of a 24-layer GPT-2 medium model reveals rank-1 attention matrices in later layers (those beyond the halfway point), indicating rank collapse. Specifically, 3 out of the last 12 later layers exhibit rank-1 attention matrices (mean rank across all the 100 runs). Our 16-layer GPT-2 medium variant, trained with Inheritune, demonstrates improved rank across all layers, highlighting the effectiveness of our approach. Notably, none of the later layers in our 16-layer variant exhibit rank-1 attention matrices.

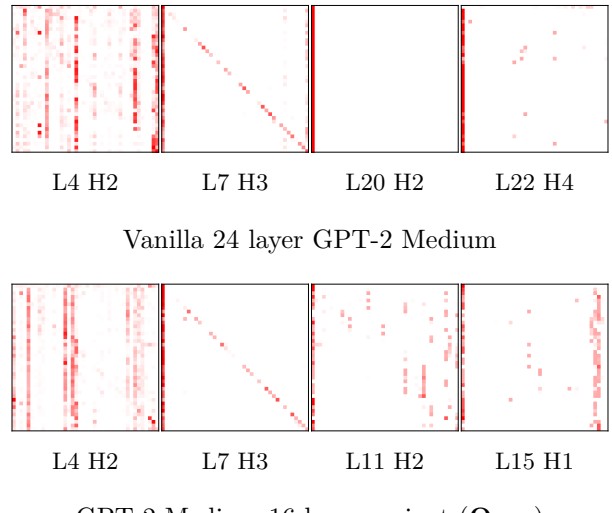

Figure 9: **Inheritune preserves effective attention patterns.** Comparison of attention patterns across layers (L) and heads (H) in two GPT-2 medium models: (top) vanilla 24-layer model trained from scratch, (bottom) 16-layer variant trained with Inheritune. Attention maps are averaged over three randomly selected strings, with 40 tokens each from the validation. Darker colors indicate higher attention scores. Inheritune maintains focused attention even in deeper layers, contrasting with the uniform patterns in the vanilla model's deeper layers.

### 4.3 Ablations and Limitations

We conducted extensive experiments to better understand which sub-module initializations within a transformer block lead to improved generalization (in terms of validation loss) and faster convergence. For these ablations, we fixed the model to a 16-layer GPT-2 Medium variant and explored three different sub-module initializations using weights from a 24-layer GPT-2 Medium reference model. We initialize the transformer

| Layers | Initialization | Steps | Pre-train Val loss ($\downarrow$) |
|--------|----------------|-------|-----------------------------------|
| 16 | Attention | 100K | 2.84 |
| 16 | MLP | 100K | 2.85 |
| 16 | Attention and MLP[*] | 100K | **2.80** |
| 16 | **Ours** | 100K | **2.81** |

Table 3: **Impact of initializing various sub-modules within a transformer block.** We compare validation loss of a 16-layer GPT-2 Medium variant when different sets of sub-modules are initialized with weights from the first 16 layers of a 24-layer GPT-2 Medium reference model. Submodules without the [*] marker also include layernorm weights. All models are trained with the OpenWebText dataset. Key findings: (1) Inheritune initialization and Attention and MLP initialization result in similar performance improvements; (2) layernorm initialization shows minimal impact. The training curves of corresponding models are presented in Figure 19. The lowest and our corresponding validation losses (lower is better) are highlighted in **bold**.

blocks with 1) Attention block (key, query, value, and projection weights) along with the layernorm[2], 2) Attention and MLP (FFN) weights without the layer-norm weights, and 3) MLP block weights along with the layernorm. We note that Inheritune performs initialization by inheriting the entire layer weight i.e. attention, MLP along with the layernorm weights.

As shown in Table 3, models initialized with both Attention and MLP weights achieve the best performance, irrespective of the LayerNorm initialization. A detailed validation loss versus training steps plot is provided in the supplementary Figure 19. These results suggest that jointly initializing the Attention and MLP submodules offers a clear advantage over initializing either component alone. Interestingly, we also find that initializing only the Attention or only the MLP weights yields comparable improvements in both convergence speed and final validation loss.

**Sensitivity Analysis for attention rank computation and lazy layers.** Next, we analyze the sensitivity of the approximate rank to the variance threshold $\tau$. Following the rank computation methodology described in Section 2, we perform rank analysis on the full GPT-2 Medium and GPT-2 Large models using a randomly sampled subset of the OpenWebText's validation set, varying $\tau \in \{0.8, 0.85, 0.9, 0.95\}$. As shown in Figure 10, the MaxRank($l$) (y-axis) remains highly stable for the *lazy layers* across all values of $\tau$ for both the models.

**Limitations of our work.** Our analysis primarily focuses on pre-layernorm (Pre-LN) architectures. By default, we initialize training at $l = L/2$, i.e. from the midpoint of the model where early layers have already developed strong representations to minimize the total number of training rounds. Figure 20 presents an ablation study showing training curves across three rounds performed following our proposed method. Notably, by the third round ($R = 3$), models trained with Inheritune match the validation loss of their reference counterpart (Full model). Despite its effectiveness, Inheritune is computationally expensive, as it may require multiple rounds of training during the growth phase. Lastly, our current analysis focuses exclusively on attention submodules; extending this framework to feedforward (MLP) layers remains an important direction for future research.

## 5 Related Works

**Attention degeneration** has been studied in the past through the lens of attention rank collapse Dong et al. (2021) leading to representation collapse, and attention entropy collapse Zhai et al. (2023) leading training instability. This also has been studied is a theoretical setup for transformer models by Noci et al. (2022); Barbero et al. (2024); Wu et al. (2024). Recently He et al. (2023) address rank collapse in self-attention networks (SANs) without residual connections or layer norms, using two different model initialization techniques that enable faithful signal propagation—i.e., $\Sigma_L$ of $A(X^L)$ does not collapse in deeper layers. However, this approach significantly slows down training. Noci et al. (2022) proposes scaling residual

---

[2]In GPT-2 models layernorm blocks are parameterized.

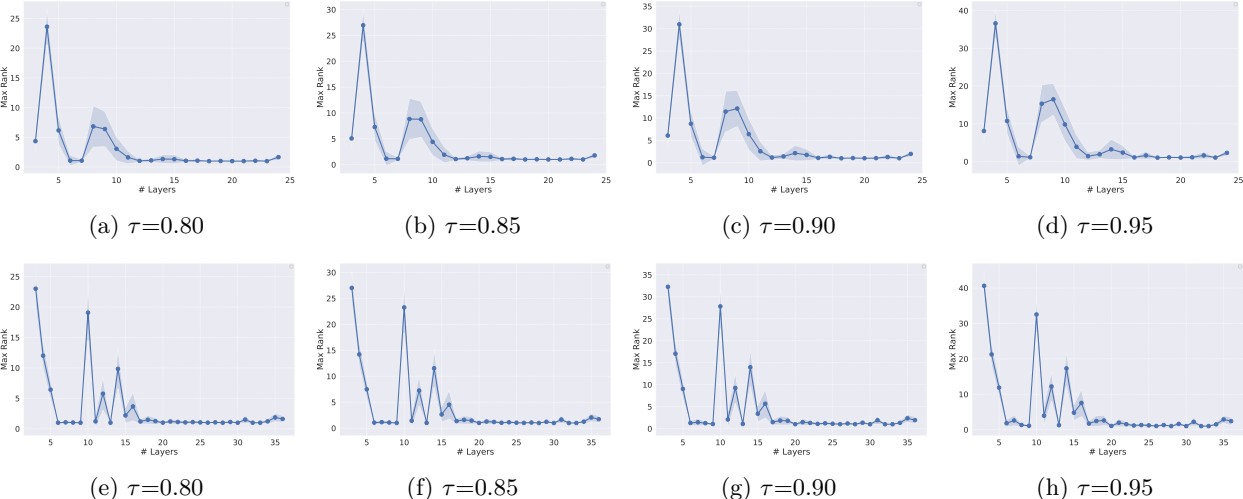

Figure 10: **Lazy layers remain robust under variations of $\tau$.** To study the sensitivity of attention rank to the variance–ratio threshold $\tau \in \{0.8, 0.85, 0.9, 0.95\}$, we visualize MaxRank($l$) as a function of layer index for (a–d) the GPT-2 Medium full model with 24 layers (top row) and (e–h) the GPT-2 Large full model with 36 layers (bottom row).

connections by $1/\sqrt{L}$, while Barbero et al. (2024) suggest that adding additional tokens to already long sequences of repeated tokens can help mitigate collapse. In contrast to prior works, we address attention degeneration by developing smaller models that eliminate structural inefficiencies and training these models to match the performance of their larger, inefficient counterparts.

**LLM training recipes and model initialization.** The stacking method Gong et al. (2019); J. Reddi et al. (2023) employs a stage-wise training strategy that uses weights from initial layers to initialize later layers has been shown to be effective for LLM training both empirically Gong et al. (2019); J. Reddi et al. (2023); Du et al. (2024) and theoretically Agarwal et al. (2024). Knowledge distillation Hinton et al. (2015) has been very successful in training small LMs. In some cases Turc et al. (2020); Sanh et al. (2019) the smaller student model is also initialized with teacher layers, though this is often done without clear explanation or intuition. Recent works in model initialization, such as Trockman & Kolter (2023), have studied synthetic attention patterns for initialization, primarily in vision settings. However, such methods have limited success in language models. Xu et al. (2024) use weight initialization for faster fine-tuning of vision models. In contrast, our proposed recipe focuses on creating smaller model by eliminating specific structural inefficiency in *lazy layers*. This distinction sets our work apart in terms of both objective and methodology.

## 6 Conclusion

In this work, we identify a structural inefficiency in deep decoder-style LLMs, which we term attention collapse, where attention matrices in deeper layers often degenerate into near rank-one structures, rendering these layers ineffective. These ineffective layers referred to as *lazy layers* contribute little to the model's representational power. To address this, we introduce Inheritune, a multi-stage training recipe that initializes a smaller model using a few potent early layers from a larger pre-trained model and then progressively trains and expands it through multiple rounds. Our experiments demonstrate that models trained with Inheritune can match or even surpass the performance of their larger counterparts despite having significantly fewer layers. By mitigating attention collapse, our approach produces compact and highly performant models, offering a new path toward designing smaller, more attentive architectures from the ground up.

## Acknowledgments

This research was supported by NSF Grant 2217069 and by UT Austin's Center for Generative AI and the Machine Learning Lab. We thank Nived Rajaraman for helpful discussions and the anonymous reviewers for their valuable feedback.

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

# Supplementary Materials

## Contents

## A  Frequently Asked Questions

### A.1  Is your method not just pruning?

Inheritune is a stage-wise efficient training recipe that addresses a structural issue in decoder-style transformer blocks—Attention collapse—which we consistently observed across multiple models.

Unlike pruning, Inheritune includes a growth phase where the model is expanded until it outperforms the reference model (refer Algorithm 1 and Figure 20). Pruning doesn't always require re-training (Ma et al., 2023), whereas Inheritune may need multiple rounds of re-training. To the best of our knowledge, no pruning method has explicitly studied or resolved attention collapse in LLMs. Our method has closer proximity to efficient training recipes employing model initialization (J. Reddi et al., 2023; Du et al., 2024) or warm starting (Ash & Adams, 2020).

### A.2  Is the comparison with baseline models trained from scratch unfair since Inheritune uses weights from pre-trained models?

In **Baseline-II** (refer Section 4) we have also compared our method with warm started baselines which also uses pre-trained model weights for model initialization for fair comparisons.

For **Baseline-I**, we include much larger reference models as well as same-sized models trained for twice as many steps. Remarkably, Inheritune still outperforms both. We believe these findings are novel and reveal a new axis for scaling.

### A.3  The attention collapse analysis is not holistic?

**TL;DR.**  We analyzed the phenomenon of attention collapse across four datasets and four different model architectures, with model sizes ranging from millions to billions of parameters.

We evaluated attention patterns and analyzed the phenomenon of attention collapse using four datasets: **OpenWebText**, **FineWeb**, **RedPajama**, and **C4**. Our analysis used two complementary metrics namely, approximate rank and approximate mass to quantify the structure of attention matrices. We conducted attention collapse analysis on a range of models, including **GPT-2** (Medium 355M, Large 770M, XLarge 1.5B), **LLaMA-3** (3B and 8B models ), **OLMo** 1B, **Cerebras-GPT** 2.7B, **Falcon** 7B and **LLaMA-1** (OpenLLaMA 3B, 7B, 13B) which features a notably different architecture. The evaluation was performed on 10K tokens (100 samples × 100 tokens each), and we also provide visualizations of the resulting attention patterns.

### A.4 What is the connection between Attention collapse and Attention sinks?

The term Attention sink (Xiao et al., 2024) refers to a phenomenon where a specific token in a sequence receives disproportionately high attention scores compared to other tokens in the attention map.

In our analysis of Attention collapse, we also observed sink-like behavior for certain tokens across all attention maps (see Figure 9 and Figure 12). However, unlike typical attention sinks, we found that beyond the sink token, **no meaningful attention structure remains**: all other tokens receive nearly uniform attention scores. We further connect this behavior to the emergence of *lazy layers*. Therefore our analysis has unique insights compared to attention sinks.

## B Extended Discussion on Attention Collapse

### B.1 Attention Collapse in LLaMA-3 models

We conducted a rank analysis on a contemporary LLaMA-3 base model with 8B parameters. We compute $\text{Rank}^{(h,l)} = \frac{1}{N} \sum_{n=1}^{N} k_{n,h,l}^*$, where $N = 100$ sequences are sampled from a subset of the C4 dataset (Raffel et al., 2019). As shown in Figure 11, we observe that nearly 50% of the attention heads (500 out of 1024 across all layers) are close to near rank-1, highlighted in red. This presents an interesting case: in very large modern architectures such as LLaMA-3 8B, while there may not be entire *lazy layers*, a substantial number of heads within many layers exhibit degeneracy. A different pattern of attention collapse compared to GPT-2 models can be attributed to the architectural differences between these models.

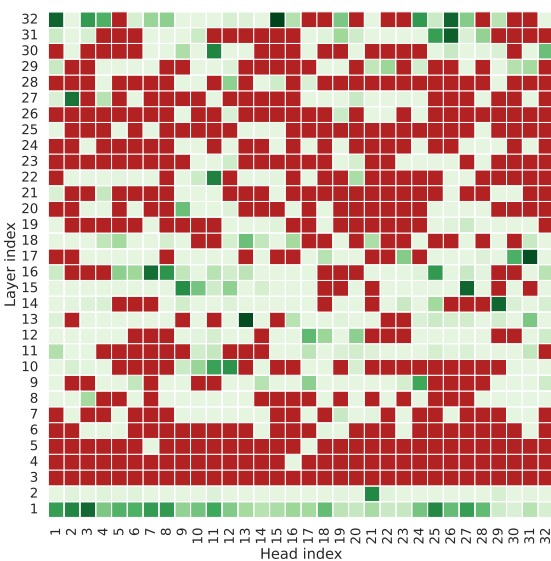

Figure 11: **Rank analysis of LLaMA-3 8B reveals that nearly half of the attention heads exhibit rank collapse.** We analyze the LLaMA-3 8B model, which contains 32 heads per layer ($32 \times 32$), using the rank metric defined in Section 2. The results are visualized as a heatmap of head index vs. layer index. Potent (non-collapsed) heads are shown in varying shades of green, where higher intensity indicates higher rank, while rank-collapsed heads (near rank-1) are highlighted in red. Approximately 50% of all attention heads exhibit rank collapse, indicating widespread degeneracy.

### B.2 Attention Pattern Visualization of Potent and Lazy Layers

Following the analysis in Section 2 and Figure 1, we present the attention patterns of two representative layers from the GPT-2 XL model: a lazy layer (Layer 30) and a potent layer (Layer 8), as shown in Figure 12. The attention patterns of these layers exhibit distinctly different behaviors. In particular, the lazy layer demonstrates a clear collapse, where attention concentrates almost exclusively on the first token. Next,

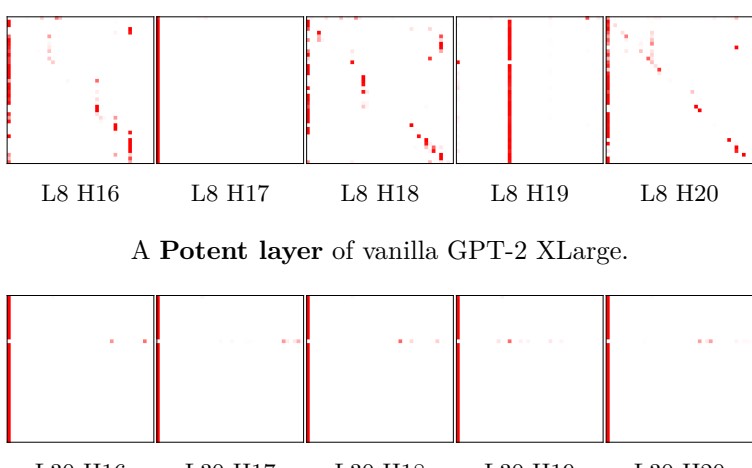

A **Potent layer** of vanilla GPT-2 XLarge.

A **lazy layer** of vanilla GPT-2 XLarge.

Figure 12: **Visualization of attention patterns in lazy and non-lazy layers of a vanilla GPT-2 XLarge model with 48 layers.** The top row displays attention patterns for various heads (H) in layer (L) 8, while the bottom row shows patterns for layer (L) 30. For visual clarity, we display the full attention maps; however, attention in GPT-2 models is inherently causal.

following the discussions in Section 4.2, we visualize the rank–layer relationship for the GPT-2 XLarge model, juxtaposing a vanilla model with a Inheritune-trained model containing half as many layers. Although both models achieve similar validation loss, the Inheritune-trained model exhibits significantly fewer *lazy layers* compared to the vanilla counterpart.

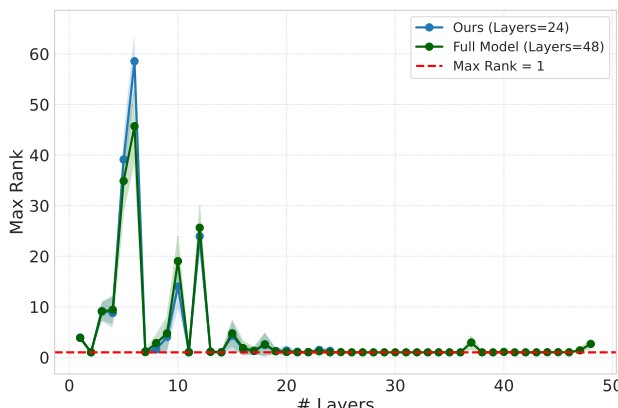

Figure 13: **Rank collapse worsens for larger LLMs, Inheritune helps to mitigate rank collapse.** The maximum (max) rank across all attention heads for each layer is plotted, following the methodology in Figure 1. We analyze a 48-layer GPT-2 XLarge model which reveals rank-1 attention matrices in later layers (those beyond the halfway point), indicating rank collapse. Specifically, 22 out of the last 24 later layers exhibit rank-1 attention matrices (mean rank across all the 100 runs). Next, Our 24-layer GPT-2 XLarge variant, trained with Inheritune, demonstrates improved rank across many layers, highlighting the effectiveness of our approach. Notably, 2 out of 12 of the later layers in our 24-layer variant exhibit rank-1 attention matrices.

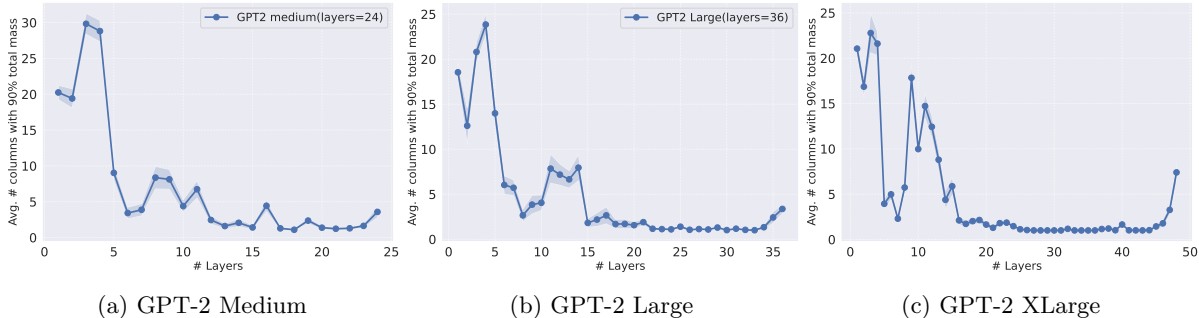

(a) GPT-2 Medium         (b) GPT-2 Large         (c) GPT-2 XLarge

Figure 14: **In decoder-style LLMs, attention matrices in deeper layers often degenerate to near single column matrices, which is a special case of near rank-1.** We compute $\text{AvgMass}^{(l)}$ (averaged over $N = 100$ randomly selected sequences each with $T = 100$ tokens) for each layer $l$ using the OpenWebText validation set. Our mass analysis of 24-layer GPT-2 Medium, 36-layer GPT-2 Large, and 48-layer GPT-2 XLarge models (L:layer, H:hidden size) reveals that attention matrices in many deeper layers collapse to single column matrices on an average.

## C    Understanding Attention Degradation using Attention Mass Analysis

In this paper, we have analyzed the attention degradation phenomenon primarily using a single metric-rank of the attention matrices (see Section 2). In this section, we aim to explore another thematically related metric to further investigate the nature of attention degradation.

We further investigated the dominant structure of the rank-1 attention matrices and observed that, on an average, many of these matrices have their mass concentrated in a single column. This intrinsic structure can be viewed as a special case of rank-1 attention matrices. To quantify this, we computed the proportion of the matrix mass contributed by each column $j$ of $A(X)$ by computing $\frac{\|A_{\cdot,j}\|_2^2}{\|A(X)\|_F^2}$, where $A_{\cdot,j}$ denotes the $j$-th column of $A(X)$, $\|A_{\cdot,j}\|_2$ is the $\ell_2$-norm of that column, and $\|A(X)\|_F$ is the Frobenius norm of $A(X)$.

Next we determine the minimal number of columns required to capture $\eta$ proportion of the total mass, formally computed as;

$$m^* = \min\left\{ m \in \{1, 2, \ldots, T\} \mid \sum_{j=1}^{m} \frac{\|A_{\cdot,j}\|_2^2}{\|A(X)\|_F^2} \geq \eta \right\},$$

Here $\eta \in (0,1)$ represents the cumulative column mass threshold. In this work, we set $\eta = 0.90$. A lower value of $m^*$ implies a stronger concentration of the attention matrix mass within fewer columns, reinforcing the phenomenon attention collapse. This analysis provides additional quantitative evidence highlighting the reduced representational capability of attention matrices in deeper transformer layers, further supporting the identification of *lazy layers*.

In Figure 14, we present the layer-wise analysis of the attention matrix mass concentration in GPT-2 models. For this analysis, (similar to the rank analysis), we computed $A(X)$ using $N = 100$ sequences selected at random from the validation set of OpenWebText (4.4M tokens), each with a sequence length of $T = 100$ tokens across all attention heads within each layer. We define the average minimal column count $m$ required to capture 90% of the attention matrix mass for each head and layer as: $m^{(h,l)} = \frac{1}{N} \sum_{n=1}^{N} m^*_{n,h,l}$. Subsequently, we aggregate this metric per layer by taking the average across all heads: $\text{AvgMass}^{(l)} = \frac{1}{H} \sum_{h'=1}^{H} m^{(h',l)}$. We observe that many of the rank-collapsed attention matrices in deeper layers exhibit single-column attention structures, as measured by the $\text{AvgMass}^{(l)}$ criterion. As shown in Figure 15, we performed a mass analysis on contemporary billion-parameter OpenLLaMA models (Geng & Liu, 2023) and observed a similar pattern of attention degradation in the deeper layers. This provides concrete evidence that the phenomenon persists across a broad range of architectures and also at the billion-parameter scale.

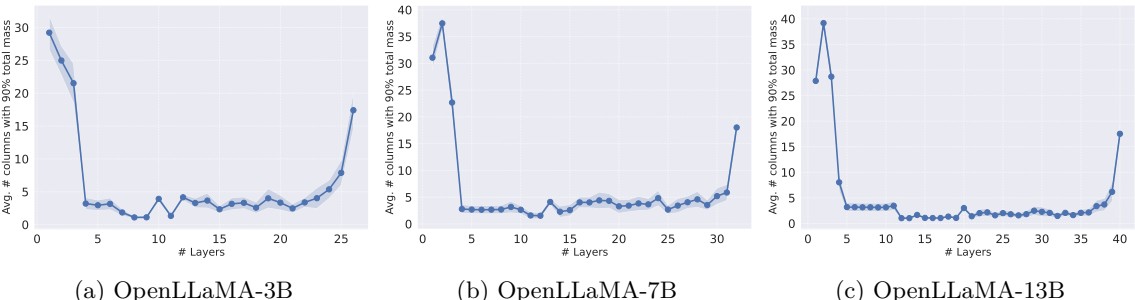

(a) OpenLLaMA-3B      (b) OpenLLaMA-7B      (c) OpenLLaMA-13B

Figure 15: **The overall mass of attention matrices in billion-scale LLMs, pre-trained on trillions of tokens, tends to concentrate in fewer columns. This phenomenon becomes increasingly pronounced as the model size grows.** We computed attention matrices using 100 tokens from a random subset of RedPajama with 1B tokens. Next, we performed 100 runs and plotted the mean and standard deviation of the mass as a function of layers for our mass analysis, respectively. We followed the same procedure as discussed in Section 2. Pre-trained checkpoints of OpenLLaMA-3B, OpenLLaMA-7B, and OpenLLaMA-13B (Geng & Liu, 2023), trained on 1T tokens from the RedPajama dataset Computer, 2023, were utilized. Overall, we observed that 90 % of the total mass of the attention matrices resides in fewer columns, with many attention matrices in the OpenLLaMA-13B model being single-column. This observation aligns closely with our analysis in Figure 1.

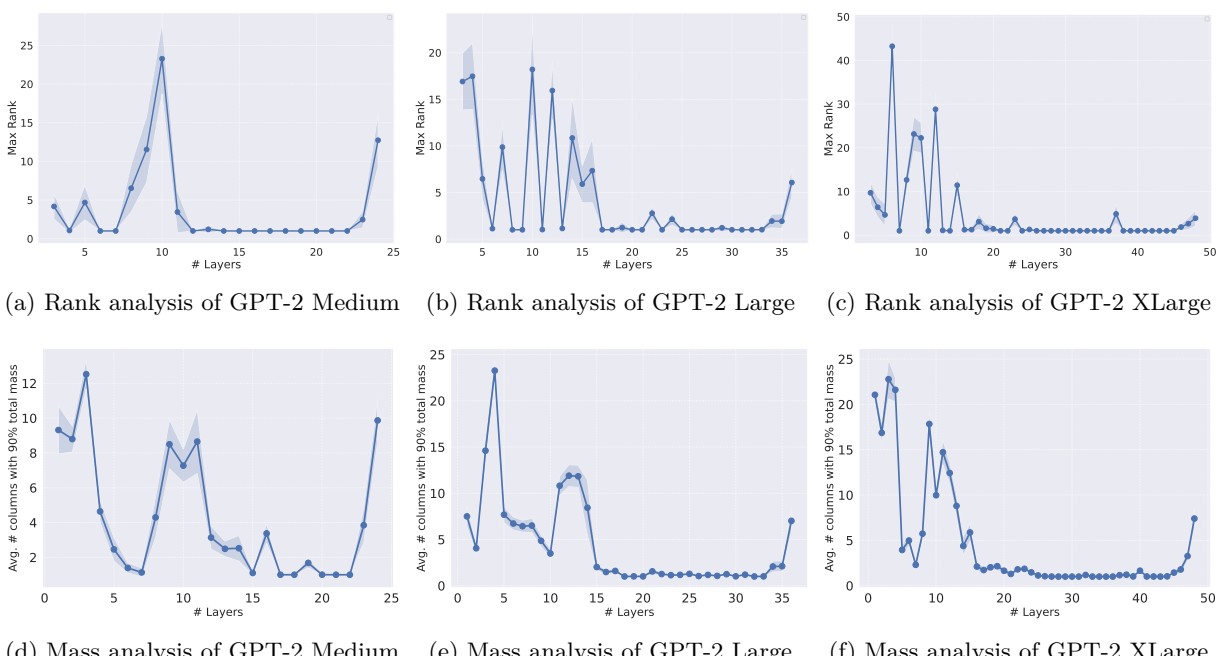

(a) Rank analysis of GPT-2 Medium    (b) Rank analysis of GPT-2 Large    (c) Rank analysis of GPT-2 XLarge

(d) Mass analysis of GPT-2 Medium    (e) Mass analysis of GPT-2 Large    (f) Mass analysis of GPT-2 XLarge

Figure 16: **In standard decoder-style LLMs, attention matrices in deeper layers often degenerate into single-column matrices, leading to layers with fully degenerated attention that fail to learn meaningful representations.** All models were trained on the OpenWebText dataset, and both rank and mass analyses were conducted using the FineWeb validation set, following the same procedure described in Figure 1. This further demonstrates the robustness of our analysis, as we reach the same conclusion using different evaluation datasets.

### C.1 Data Robustness of Attention Degeneration Analysis

In Section 2 (Figure 1), we performed a rank analysis on three pre-trained GPT-2 models—Medium, Large, and XLarge using the validation set of the OpenWebText dataset, whose training split was originally used for pre-training these models. Here, we evaluate the data robustness of our analysis by repeating the same procedure on a validation set from FineWeb, a newer and distinct dataset. Except for the dataset substitution, all experimental steps remain identical to those described earlier. The results in Figure 16 consistently show that attention tends to lose rank, particularly in deeper layers, often collapsing into near single-column structures across all models. These findings further reinforce the robustness and generality of our observations. Moreover, for the LLaMA-3 model (Figure 11) and the OpenLLaMA models (Figure 15), we used publicly available model weights and conducted our analyses on off-the-shelf datasets that were not part of the models' original training corpora.

## D    Baselines

We compare Inheritune against several baselines. While some baseline methods are illustrated using GPT-2 Large or medium (for the knowledge distillation baseline) as an example, the same methodology is consistently applied across all model variants.

**Baselines trained from scratch (random initializations) :**  We compare our Inheritune-derived model against much larger GPT-2 reference models trained from scratch for the same number of steps and similar-sized models trained from scratch for both the same and double the number of training steps.

**Baselines trained with warm started training methods.**  Here we compare our model derived using Inheritune, to similar sized models trained with various model initialization and effcient training techniques which requires model to be initialized with trained weights such as stacking, hybrid stacking, and half-width. We explain these baseline training recipes using GPT-2 Large and its variants as an example and apply the same process for other models.

**Stacking** Gong et al. (2019); J. Reddi et al. (2023) is a model initialization and efficient (stage-wise) training recipe. We train a 9-layer GPT-2 Large variant from scratch for 100K steps, then expanded the model to 18 layers by copying the weights from layers 0-8 to layers 9-17. Finally we re-trained this new 18-layer GPT-2 Large variant, using stacking initialization for an additional 100K steps.

**Hybrid stacking**: Hybrid stacking is stacking but utilizes a large pre-trained reference model for initialization instead of using its own pre-trained weights. We took the weights of layers 0-8 from the reference 36-layer GPT-2 Large model and expanded it to a 18-layer model by copying the weights to layers 0-17. We then trained this new 18-layer GPT-2 variant for 100K steps.

**Half width**: We initialized the baseline GPT-2 Large variant across the width dimension and preserved the entire depth. We copied the weights of the first half the attention heads (0-9) and MLPs of the GPT-2 Large reference model into baseline GPT-2 variant with half the width but all layers.

**Baselines trained with Knowledge Distillation**  As a baseline, we first apply logit-based knowledge distillation Hinton et al. (2015) to train a 16-layer GPT-2 Medium variant (student) initialized randomly. For the second baseline, we use a DistillBERT-style approach Sanh et al. (2019), where the student model 0-11 layers are initialized with every alternate block of its teacher, and the remaining 4 blocks are initialized using layers 18, 19, 20, and 21 of the teacher. Both baselines are trained for 50K steps, using a vanilla 24-layer GPT-2 Medium model as the teacher (our reference model).

## E    Supplementary Experiments

We provide supplementary tables and plots corresponding to the results discussed in the main paper, along with additional experiments, in this section. The final validation losses shown in Figure 2 are presented in Table 4. Similarly, the final validation losses from the training curves in Figure 5 and Figure 6 are summarized

| Layers | Initialization | AvgRank | Pre-train Val Loss (↓) |
|---|---|---|---|
| 4 | rand | N/A | 3.25 |
| 4 | 1-4 layers from vanilla GPT2 | 8.40 | 3.22 |
| 4 | 5-8 layers from vanilla GPT2 | 9.48 | 3.19 |
| 4 | 9-12 layers (lazy layers) from GPT2 | 1.22 | 3.23 |

Table 4: **Impact of initialization strategies on GPT-2 small variants.** We analyzed the rank characteristics of a vanilla GPT2-small model (125M, 12 layers) trained on OpenWebText for 100K steps. Four-layer GPT2-small variants were initialized using the first 4 layers [1–4], middle 4 layers [5–8], last 4 layers [9–12], or with random initialization, and then trained for 100K steps on OpenWebText. Models initialized with the last 4 layers performed similarly to random initialization, while those initialized with layers exhibiting higher average max ranks achieved the best validation loss, regardless of proximity to the embedding layer. The training plots and rank analysis are provided in Figure 2.

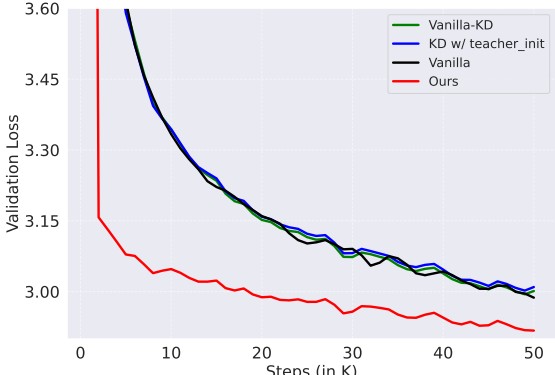

Figure 17: **A 16-layer GPT-2 Medium variant derived using** Inheritune **converges faster and generalizes better than a same-sized model trained with logit-based distillation baselines.**

in Table 5. We also include training curves for models trained using various warm-started baselines (i.e., models initialized with learned weights) compared against our method, these results correspond to Table 2 and are visualized in Figure 18. Finally, Figure 19 presents the training curves for models used in the ablation study discussed in Section 4.3.

**Distillation vs. Inheritune.** We conducted an additional experiment to compare Inheritune with knowledge distillation as a baseline. Specifically, we trained GPT-2 Medium variants with 16 layers under three different settings. First, we performed logit-based distillation Hinton et al. (2015), transferring knowledge from a 24-layer vanilla GPT-2 Medium (teacher) to a 16-layer student model. Second, we applied a DistilBERT-style distillation Sanh et al. (2019), where the student is initialized with the teacher's layers. Finally, we trained a 16-layer GPT-2 Medium model from scratch using vanilla training. Across all comparisons, the model trained with our Inheritune recipe outperformed both distilled variants, achieving faster convergence and substantially better generalization after 50K training steps. We defer a thorough investigation of the relationship between Inheritune and distillation-based approaches to future work. The training configurations are provided in Section F.

# F   Architectural and Training Details

## F.1   GPT-2 Model configurations

For our main experiments, we focus on three sizes of GPT-2 models Radford et al. (2019): GPT-2 XLarge with 1.5B parameters, GPT-2 Large with 770M parameters, and GPT-2 Medium with 355M parameters. We

| Models | Layers | Initialization | Steps | Pre-train Val loss (↓) |
|---|---|---|---|---|
| GPT-2 Medium | 24 | rand init | 100K | **2.81** |
| | 16 | rand init | 100K | 2.86 |
| | 16 | rand init | 200K | 2.83 |
| | 12 | Ours | 100K | 2.87 |
| | 14 | Ours | 100K | 2.84 |
| Final Model ⟶ | 16 | **Ours** | 100K | **2.81** |
| GPT-2 Large | 36 | rand init | 100K | 2.85 |
| | 18 | rand init | 100K | 2.97 |
| | 18 | rand init | 200K | 2.84 |
| | 18 | **Ours** | 100K | **2.80** |
| GPT-2 XLarge | 48 | rand init | 100K | 2.65 |
| | 24 | rand init | 100K | 2.69 |
| | 24 | rand init | 200K | **2.62** |
| | 24 | **Ours** | 100K | **2.64** |

Table 5: **Inheritune achieves superior performance with reduced model size.** Comparison of Inheritune-trained models (24-layer GPT-2 XLarge, 18-layer GPT-2 Large, and 16-layer GPT-2 Medium) against full-sized counterparts and extended training baselines. The training steps of two different baselines are reported in the table, we use validation loss on the OpenWebText validation set. Note: GPT-2 Large and XLarge uses one round of Inheritune; GPT-2 Medium uses three rounds. The lowest and our corresponding validation losses (lower is better) are highlighted in **bold**.

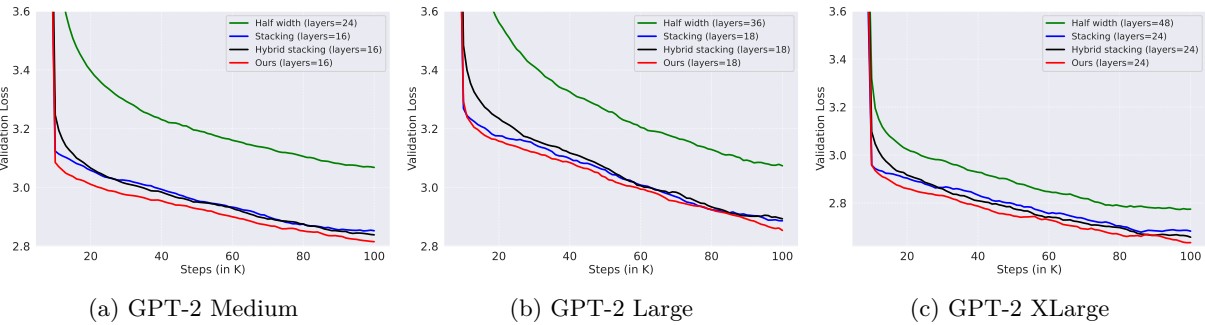

(a) GPT-2 Medium          (b) GPT-2 Large          (c) GPT-2 XLarge

Figure 18: **Models derived using Inheritune outperform three warm-started baselines (Baseline-II) in terms of final validation loss.** Our models demonstrate better convergence and generalization compared to all baselines. All the models are trained with OpenWebText for 100K steps. The curves are smoothed for visual clarity.

developed several variants of these models by adjusting the number of layers, i.e., reducing the depth for vanilla models for to be trained with Inheritune and baseline mathods. In one baseline namely, the half-width variant we modified both the hidden size (and consequently, the number of attention heads) in addition to the depth, as shown in Figure 2. The key architectural configurations of the reference, proposed, and baseline models discussed in this paper are summarized in Table 6.

## F.2 Training details of GPT-2 models

All GPT-2 models used in this study (unless otherwise stated) were pre-trained on the OpenWebText dataset, which contains approximately 10B tokens. We employed a dataloader that samples tokens with replacement, meaning that the tokens used for training are not necessarily unique, following the approach of Liu et al.

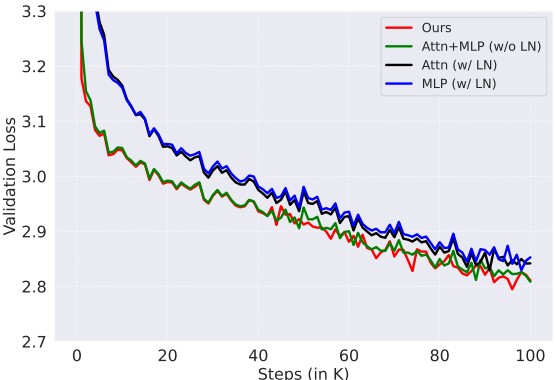

Figure 19: **Full training curves of 16-layer GPT-2 variants trained during ablations.** We analyze Inheritune approach while initializing some specific sub-modules in transformer blocks. Here, we initialize each transformer block of a 16-layer GPT-2 Medium variant with three different configurations. First, we separately initialize attention and MLPs (FFNs) submodules; second, we initialize the attention and MLP weights while randomly initializing the layer norms. Finally, we perform Inheritune-initialize only the attention and MLP weights with all the respective layer norms.

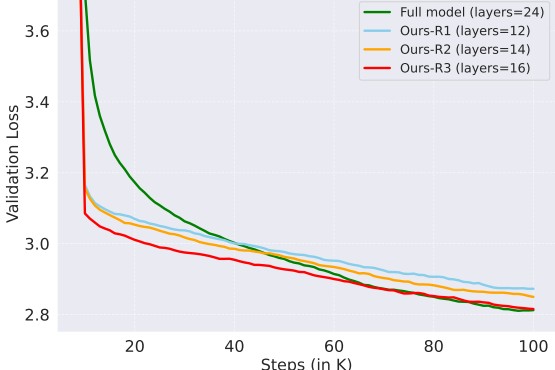

Figure 20: **Training curves for the 24-layer GPT-2 Medium model (full model) and three rounds of training following Inheritune recipe (to grow the model)**. We present the training trajectories for all GPT-2 Medium variants trained using the Inheritune recipe. The final model obtained after the third round ($R = 3$) with $L = 16$ layers matches the final validation loss of the full model. All models are trained for 100K steps on the OpenWebText dataset.

| Model Family | Type | Layers | Hidden Size | Heads | Notes |
|---|---|---|---|---|---|
| GPT-2 XLarge (1.5B) | Reference | 48 | 1600 | 25 | Original architecture |
| | Variants | 24 | 1600 | 25 | Reduced-depth |
| GPT-2 Large (770M) | Reference | 36 | 1280 | 20 | Original architecture |
| | Variants | 18 | 1280 | 20 | Reduced depth |
| GPT-2 Large$^{\dagger}$ (668M) | Reference | 32 | 1280 | 20 | Custom architecture |
| | Variants | 16 | 1280 | 20 | Reduced depth |
| GPT-2 Medium (355M) | Reference | 24 | 1024 | 16 | Original architecture |
| | Variants | 16 | 1024 | 16 | Reduced depth |
| GPT-2 Small (125M) | Reference | 12 | 768 | 12 | Original architecture |
| | Variants | 4 | 768 | 12 | Reduced depth |

Table 6: **Overview of all GPT-2 models used in this study and their architectural configurations.** GPT-2 models are Pre-LN based architectures. The model configurations employed for the stacking and hybrid stacking baselines are identical to those of our variants. For the half-width baseline, we used GPT-2 variants with half the hidden size and number of attention heads.

(2023). For evaluating the pre-trained models, we used the validation split of the same dataset, which contains 4.4M tokens. The sole exception to this setup is the GPT-2 models trained on the FineWeb edu with 10B tokens (Figure 7), where we used unique tokens for training by employing a dataloader where we sample without replacement.

We employed the AdamW optimizer with $\beta_1 = 0.90$ and $\beta_2 = 0.95$. All GPT-2 models were trained on a single NVIDIA A100 GPU (40 GB memory) with gradient accumulation. For the GPT-2 XLarge and its variants, we utilized an NVIDIA H100 GPU. Most hyperparameters were adapted from Liu et al. (2023), with key details discussed in this section.

**Hyper-parameter details of GPT-2 Medium and variants.**

- Batch size: 394K tokens
- Learning rate: $3 \times 10^{-4}$
- Warmup steps: 2K
- Scheduler type: cosine decayed to $1 \times 10^{-5}$
- Weight decay: 0.1
- Gradient clipping value: 1
- Total training steps: 100K

**Hyper-parameter details of GPT-2 Large and variants.**

- Batch size: 128K tokens
- Learning rate: $2 \times 10^{-4}$
- Warmup steps: 2K
- Scheduler type: cosine decayed to $1 \times 10^{-5}$
- Weight decay: 0.1
- Gradient clipping value: 1
- Total training steps: 100K

**Hyper-parameter details of GPT-2 XLarge and variants.**

- Batch size: 128K tokens

- Learning rate: $1.5 \times 10^{-4}$

- Warmup steps: 2K

- Scheduler type: cosine decayed to $1 \times 10^{-5}$

- Weight decay: 0.1

- Gradient clipping value: 1

- Total training steps: 100K

**Hyper-parameter details of knowledge distillation training.**

We use the below loss for as our distillation based training loss. The validation loss is the student_loss.

$$\text{Total\_loss} = \alpha \cdot \text{student\_loss} + (1 - \alpha) \cdot \text{distillation\_loss}$$

- Model: 16-layer GPT-2 Medium variants

- $\alpha$: 0.6

- Batch size: 394K tokens

- Learning rate: $3 \times 10^{-4}$

- Warmup steps: 2K

- Scheduler type: cosine decay to $\frac{1}{10}$ of max learning rate

- Weight decay: 0.1

- Gradient clipping value: 1

- Total training steps: 50K

# G    Additional Experiments and Discussions

## G.1    Additional Downstream Evaluation

We extend our evaluation to models trained on OpenWebText, as described in Section 4 (see Table 7). For this analysis, we focus on the largest model considered in this work, GPT-2 XLarge, along with its variants. Downstream evaluation is performed on all datasets listed in Section 4, with the addition of Winogrande (Sakaguchi et al., 2020), BoolQ (Clark et al., 2019), and Wikitext (Merity et al., 2016). Note that the LAMBADA dataset is evaluated under two settings: missing-word prediction (accuracy) and language modeling (perplexity), and is therefore reported twice. All evaluations are implemented using the widely adopted *lm-eval-harness*.

| Task | Full model | Ours |
|------|------------|------|
| **Accuracy-based tasks** ($\uparrow$) | | |
| ARC-E | $50.38 \pm 1.03$ | $51.22 \pm 1.03$ |
| PIQA | $66.70 \pm 1.10$ | $66.87 \pm 1.10$ |
| SciQ | $77.00 \pm 1.33$ | $79.20 \pm 1.28$ |
| HellaSwag | $33.65 \pm 0.47$ | $34.20 \pm 0.47$ |
| LAMBADA | $39.90 \pm 0.68$ | $43.30 \pm 0.69$ |
| WinoGrande | $51.93 \pm 1.40$ | $53.28 \pm 1.40$ |
| BoolQ | $57.86 \pm 0.86$ | $60.40 \pm 0.86$ |
| **Average** | 53.92 | **55.50** |
| **Perplexity-based tasks** ($\downarrow$) | | |
| Wikitext | 25.46 | 25.52 |
| LAMBADA | 20.24 | 16.51 |
| **Average** | 22.85 | **21.01** |

Table 7: **Downstream evaluation of GPT-2 XLarge (1.5B) trained from scratch vs. a 24-layer model trained with Inheritune (Ours).** We evaluate both models on 7 accuracy-based tasks (higher is better) and 2 perplexity-based tasks (lower is better). All models are trained on OpenWebText. Despite using half the depth, the Inheritune model performs favorably compared to the full-sized counterpart. Best average scores are highlighted in **bold**.

### G.2    Additional Evidence of Attention Collapse in Modern LLMs

Following the discussion in Section B.1, we extend our analysis to several additional open-weight base LLMs (see Figures 21 and 22), including Falcon-7B[3], OLMo-1B[4], Cerebras-GPT-2.7B[5], and LLaMA-3-3B[6].

Many of these models incorporate modern architectural components, including Grouped Query Attention (GQA), RoPE positional embeddings, and RMS normalization applied both before and after the attention module and trained with billions-trillions of tokens. We visualized the attention heads of each model using heatmaps of head index versus layer index. Across all models, we observe a predominance of near rank-1 attention heads, indicating widespread attention collapse.

### G.3    Additional Results on the Functional Ineffectiveness of Lazy Layers

Following the discussion in Section 2.1 on the poor transferability of lazy layers, we conduct an additional experiment. We first train a 24-layer GPT-2 Medium model (reference) on OpenWebText for 100K steps. We then train a second model of identical size using the same data and training configuration. In this model, the deeper layers (layers 16–24) are initialized with the corresponding weights from the reference model, while the remaining layers are randomly initialized. As shown in Figure 23, reusing lazy layers consistently degrades performance relative to training all layers from scratch. This result complements our findings in Section 2.1 and provides further evidence that lazy layers do not encode transferable or reusable representations suitable for initialization. We used a batch size of 50K tokens with 2 H100 GPUs; all other training details remain unchanged (see Section F.2)

---

[3] `https://huggingface.co/tiiuae/falcon-7b`
[4] `https://huggingface.co/amd/AMD-OLMo`
[5] `https://huggingface.co/cerebras/Cerebras-GPT-2.7B`
[6] `https://huggingface.co/meta-llama/Llama-3.2-3B`

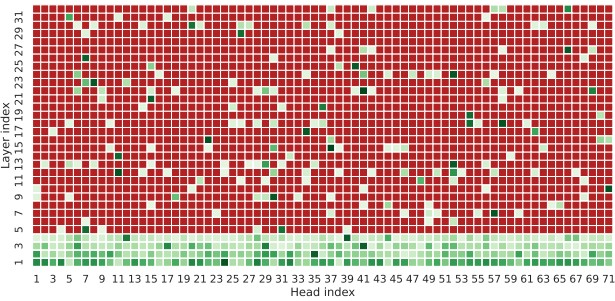

Figure 21: **Rank analysis of the Falcon-7B model reveals widespread attention collapse.** We analyzed Falcon-7B (version 1), which comprises 31 layers with 71 attention heads per layer, using the rank metric defined in Section 2. The results are visualized as a heatmap with layer index on one axis and head index on the other. Potent (non-collapsed) heads are shown in varying shades of green, with higher intensity indicating higher rank, while rank-collapsed (near rank-1) heads are highlighted in red. Overall, approximately 75% of attention heads exhibit rank collapse, indicating substantial degeneracy across the model.

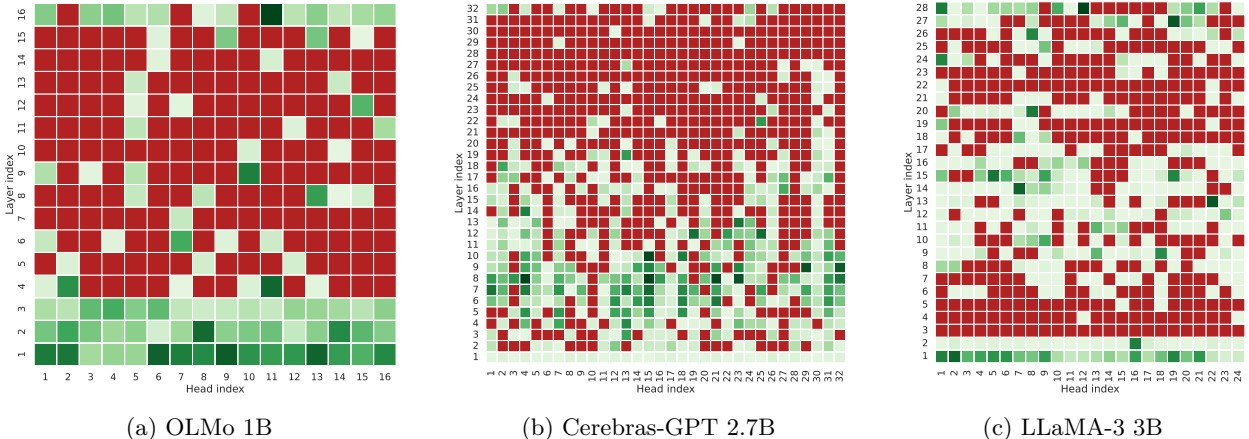

(a) OLMo 1B      (b) Cerebras-GPT 2.7B      (c) LLaMA-3 3B

Figure 22: **Rank analysis of large open-weight LLMs reveals widespread attention collapse.** We analyze three base open-weight language models using the rank metric defined in Section 2. (a) OLMo (1B) exhibits rank collapse in approximately 58% of attention heads; (b) Cerebras-GPT (2.7B) shows a similar level of collapse at roughly 58%; and (c) LLaMA-3 (3B) exhibits rank collapse in about 50% of attention heads. Each heatmap plots attention head index versus layer index. Potent (non-collapsed) heads are shown in varying shades of green, with higher intensity indicating higher rank, while rank-collapsed (near rank-1) heads are highlighted in red.

### G.4 Additional Training Results in the Extended Training Regime

In this section we present a supplementary result where we have extended the training steps from 100K (as discussed in Section 4) to 200K steps to gauge the potential Inheritune holds for longer training runs. As shown in Figure 24 a GPT-2 medium (24 layer full model) is compared against a 18 layer GPT-2 medium variant trained following Inheritune recipe. We have inherited $l = 18$ layers based on the previously known best configuration for GPT-2 medium variant where a 16 layer model matches the performance its full sized counterpart. We observe that the Inheritune model with 18 layers achieves a lower final validation loss (2.86) than the 24-layer baseline (2.87), demonstrating that Inheritune continues to provide benefits even with extended training. We used a batch size of 50K tokens with 2 H100 GPUs; all other training details remain unchanged (see Section F.2).

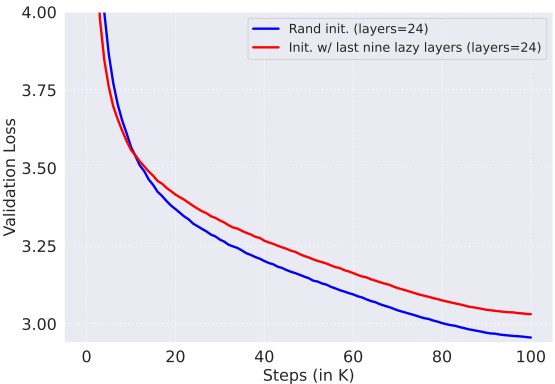

Figure 23: **Initializing deeper layers of GPT-2 Medium with lazy layers degrades performance.** We first train a full GPT-2 Medium model from scratch on OpenWebText for 100K steps. We then construct a second model in which layers 16–24 are initialized using the corresponding *lazy layers* layers from the trained reference model, while layers 1–15 are randomly initialized. Both models are trained using the same optimization and data settings. The model whose deeper layers are initialized with lazy layers performs significantly worse than its counterpart trained entirely from scratch.

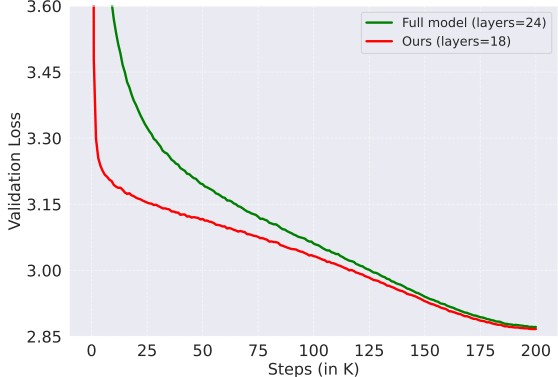

Figure 24: **Inheritune continues to provide gains under extended training.** Comparison between a full 24-layer GPT-2 Medium model and an 18-layer GPT-2 Medium variant trained using the Inheritune recipe, both trained on OpenWebText for 200K steps. The Inheritune model achieves a lower final validation loss (2.86) than the full 24-layer baseline (2.87).

### G.5  Discussion on Lazy Layers

In this section, we provide a formal characterization of *lazy layers* through the rank-based criterion.

**Definition G.1** (**Lazy Layer (Rank-based)**)**.** Let $\{X_n\}_{n=1}^N$ be a collection of $N$ sequences, each consisting of $T$ tokens. For each layer $l \in \{1, \dots, L\}$ and head $h \in \{1, \dots, H\}$, let $A_{n,h,l} \in \mathbb{R}^{T \times T}$ denote the attention matrix produced by head $h$ in layer $l$ on input sequence $X_n$.

Let $\sigma_1(A_{n,h,l}) \geq \cdots \geq \sigma_T(A_{n,h,l}) \geq 0$ be the singular values of $A_{n,h,l}$. For a threshold $\tau \in (0, 1)$, we define the approximate rank as

$$k_{n,h,l}^*(\tau) \;=\; \min\left\{ k \in \{1, \dots, T\} \;\middle|\; \frac{\sum_{i=1}^k \sigma_i(A_{n,h,l})^2}{\sum_{j=1}^T \sigma_j(A_{n,h,l})^2} \geq \tau \right\}. \tag{2}$$

We choose $\tau$ via an ablation study; throughout the paper we use $\tau = 0.90$ (see Section 4.3).

Define the head-wise aggregated rank and layer-wise aggregated rank as

$$\text{Rank}(h,l) \;=\; \frac{1}{N}\sum_{n=1}^{N} k_{n,h,l}^{*}(\tau), \qquad \text{MaxRank}(l) \;=\; \max_{h\in\{1,\ldots,H\}} \text{Rank}(h,l). \tag{3}$$

For a group of $L$ layers (where $L$ denotes the number of contiguous layers in the group), define

$$\text{AvgRank} \;=\; \frac{1}{L}\sum_{l=1}^{L} \text{MaxRank}(l). \tag{4}$$

> **Lazy Layer.** A layer $l$ is a **Lazy Layer** if $\text{MaxRank}(l) = 1$.
>
> **Lazy Layers.** A group of contiguous blocks of $L$ layers is termed **Lazy** if $\lfloor \text{AvgRank} \rfloor = 1$.

## H Theoretical Analysis

### H.1 Causal Attention Rank-1 Collapse and Vanishing Gradients

We isolate a simple mechanism in decoder-style (causal) self-attention: when a head becomes highly concentrated on a single sink position across time (e.g., the first/BOS token), the resulting attention matrix is approximately rank-1, and the corresponding gradients through the softmax saturate, yielding vanishing updates to $W_Q$ and $W_K$.

**Setup.** We inherit notations discussed in Section 2. Let $X \in \mathbb{R}^{T\times d}$ denote a sequence of $T$ tokens. Define $Q = XW_Q$, $K = XW_K$, $V = XW_V \in \mathbb{R}^{T\times d}$ and masked logits

$$\tilde{A} = \frac{1}{\sqrt{d}} QK^\top + M, \qquad M_{ij} = \begin{cases} 0 & i \geq j, \\ -\infty & i < j, \end{cases} \tag{5}$$

with attention weights $A = \text{softmax}(\tilde{A})$ applied row-wise over unmasked indices, and the final output becomes $O = AV$. Let $\mathcal{L}$ denote the training objective.

**Definition (attention collapse).** Let $A \in \mathbb{R}^{T\times T}$ be an attention matrix (row-stochastic: $A_{ij} \geq 0$ and $\sum_{j=1}^{T} A_{ij} = 1$ for each row $i$). Fix an index $j^\star \in \{1,\ldots,T\}$ that is unmasked for all rows (e.g., $j^\star = 1$ for the BOS token). We say that $A$ is $\varepsilon$-sink-collapsed to $j^\star$ if, for every row $i \in \{1,\ldots,T\}$,

$$\sum_{j\neq j^\star} A_{ij} \;\leq\; \varepsilon. \tag{6}$$

Equivalently, each row places at least $1 - \varepsilon$ probability mass on the same column $j^\star$, i.e.,

$$A_{ij^\star} \;=\; 1 - \sum_{j\neq j^\star} A_{ij} \;\geq\; 1 - \varepsilon, \qquad \forall i \in \{1,\ldots,T\}. \tag{7}$$

**Theorem H.1** (Attention collapse to rank-1 sink and vanishing gradients). *Consider a single-head causal self-attention module with $A = \text{softmax}(\frac{1}{\sqrt{d}}QK^\top + M)$ and $O = AV$. If $A$ is $\varepsilon$-sink-collapsed to some $j^\star$ as in equation 6, then:*

*(i) **Rank-1 collapse.** Let $A_0 \triangleq \mathbf{1}e_{j^\star}^\top$ (rank-1). Then*

$$\|A - A_0\|_F \leq \varepsilon\sqrt{2T} \;\Rightarrow\; \sigma_2(A) \leq \|A - A_0\|_2 \leq \|A - A_0\|_F \leq \varepsilon\sqrt{2T}, \tag{8}$$

*so $A$ is numerically rank-1 where $\varepsilon > 0$ is small.*

***(ii) Vanishing gradients through softmax.*** *For each row $i$, letting $a^{(i)}$ be the entries of that row (restricted to unmasked indices),*

$$\frac{\partial \mathcal{L}}{\partial \tilde{a}^{(i)}} = \left( \mathrm{diag}(a^{(i)}) - a^{(i)} a^{(i)\top} \right) \frac{\partial \mathcal{L}}{\partial a^{(i)}}, \quad \left\| \frac{\partial \mathcal{L}}{\partial \tilde{a}^{(i)}} \right\|_2 \le 2\varepsilon \left\| \frac{\partial \mathcal{L}}{\partial a^{(i)}} \right\|_2. \tag{9}$$

*Thus, once rows are near one-hot ($\varepsilon \to 0$), the gradient to logits vanishes.*

***(iii) Vanishing gradients to $W_Q$ and $W_K$.*** *Moreover,*

$$\left\| \frac{\partial \mathcal{L}}{\partial W_Q} \right\|_F \le \frac{2\varepsilon}{\sqrt{d}} \|X\|_2 \|K\|_2 \left\| \frac{\partial \mathcal{L}}{\partial A} \right\|_F, \quad \left\| \frac{\partial \mathcal{L}}{\partial W_K} \right\|_F \le \frac{2\varepsilon}{\sqrt{d}} \|X\|_2 \|Q\|_2 \left\| \frac{\partial \mathcal{L}}{\partial A} \right\|_F, \tag{10}$$

*so parameter updates to queries/keys shrink linearly with the collapse level $\varepsilon$.*

*Proof.* We prove the three claims sequentially.

**(i)** Let $A_0 = \mathbf{1} e_{j^\star}^\top \in \mathbb{R}^{T \times T}$. Clearly $A_0$ has rank 1. As the Frobenius norm is the $\ell_2$ norm aggregated over rows, it follows that $\|A - A_0\|_F^2 = \sum_{i=1}^{T} \|a^{(i)} - e_{j^\star}\|_2^2$. Fix a row $a \in \mathbb{R}^T$ (without loss of generality we omit the superscript $i$), under $\varepsilon$-sink-collapse, $\sum_{j \ne j^\star} a_j \le \varepsilon$. Since $A$ is an attention matrix (row-stochastic), then $\sum_{j=1}^{T} a_j = 1$ and $a_j \ge 0$, hence $a_{j^\star} = 1 - \sum_{j \ne j^\star} a_j \ge 1 - \varepsilon$. Now consider the squared error $\|a - e_{j^\star}\|_2^2 = (a_{j^\star} - 1)^2 + \sum_{j \ne j^\star} a_j^2$. Here, $(a_{j^\star} - 1)^2 \le \varepsilon^2$ (follows from $a_{j^\star} \ge 1 - \varepsilon$), and $\sum_{j \ne j^\star} a_j^2 \le \left( \sum_{j \ne j^\star} a_j \right)^2 \le \varepsilon^2$. Thus, the squared error $\|a - e_{j^\star}\|_2^2 \le 2\varepsilon^2$. Summing over $T$ rows, gives

$$\|A - A_0\|_F^2 = \sum_{i=1}^{T} \|a^{(i)} - e_{j^\star}\|_2^2 \le \sum_{i=1}^{T} 2\varepsilon^2 = 2T\varepsilon^2 \Rightarrow \|A - A_0\|_F \le \varepsilon\sqrt{2T},$$

Hence, $\sigma_2(A) \le \|A - A_0\|_2 \le \|A - A_0\|_F \le \varepsilon\sqrt{2T}$ (following Weyl's inequality).

**(ii)** (Gradients are taken only over unmasked logits; masked positions have zero gradient.) Let $g \triangleq \partial \mathcal{L} / \partial a^{(i)}$ and $\tilde{g} \triangleq \partial \mathcal{L} / \partial \tilde{a}^{(i)}$. For the row-wise softmax, the Jacobian is $J = \mathrm{diag}(a) - aa^\top$, hence $\tilde{g} = Jg$. Since $J \succeq 0$, $\|J\|_2 \le \mathrm{tr}(J)$. Moreover, $\mathrm{tr}(J) = \sum_{i=1}^{T} a_i(1 - a_i) = 1 - \|a\|_2^2$. Under $\varepsilon$-sink collapse, $a_{j^\star} \ge 1 - \varepsilon$, so $\|a\|_2^2 \ge (1 - \varepsilon)^2$ and therefore

$$\|J\|_2 \le 1 - (1 - \varepsilon)^2 \le 2\varepsilon.$$

Thus $\|\tilde{g}\|_2 \le \|J\|_2 \|g\|_2 \le 2\varepsilon \|g\|_2$, proving equation 9.

**(iii)** We focus on $W_Q$; the argument for $W_K$ is analogous. By the chain rule, $\frac{\partial \mathcal{L}}{\partial W_Q} = X^\top \frac{\partial \mathcal{L}}{\partial Q}$. Let $\bar{G} \triangleq \frac{\partial \mathcal{L}}{\partial \tilde{A}}$. Since $\tilde{A} = \frac{1}{\sqrt{d}} QK^\top + M$, we have $\frac{\partial \mathcal{L}}{\partial Q} = \frac{1}{\sqrt{d}} \bar{G} K$. Bounding norms and using $\|X^\top H\|_F \le \|X\|_2 \|H\|_F$, (by definition)

$$\left\| \frac{\partial \mathcal{L}}{\partial W_Q} \right\|_F \le \|X\|_2 \left\| \frac{\partial \mathcal{L}}{\partial Q} \right\|_F \le \frac{1}{\sqrt{d}} \|X\|_2 \|\bar{G}\|_F \|K\|_2.$$

From (ii), row-wise $\|\partial \mathcal{L} / \partial \tilde{a}^{(i)}\|_2 \le 2\varepsilon \|\partial \mathcal{L} / \partial a^{(i)}\|_2$, and therefore $\|\bar{G}\|_F \le 2\varepsilon \|\frac{\partial \mathcal{L}}{\partial A}\|_F$. Substituting gives equation 10. $\square$

**Interpretation and training-time implication.** Theorem H.1 connects **rank-1 causal attention collapse** (sink behavior) and **vanishing gradients**: once a head becomes near-deterministic, the softmax Jacobian saturates and the head receives negligible learning signal to recover. Consequently, collapse becomes sticky (hard to recover from) after entering the collapsed regime, subsequent gradient updates tend to be too small to move the head back to a higher rank (non-collapsed) state. Over training, this stickiness can accumulate across layers especially in deeper layers which are more prone to vanishing gradients leading to increasingly persistent rank collapsed attention patterns and progressively smaller query/key weight updates.

