# OpenReview forum: "When Attention Collapses: How Degenerate Layers in LLMs Enable Smaller, Stronger Models"
_TMLR — Accepted by TMLR_

### Review · Reviewer_sKfL · 2025-12-09

**Summary Of Contributions:**

This work has a few key contributions.
Author(s) define the term attention collapse and show its prevelance in the deeper layers of decoder LLMs. Secondly, they propose Inheritune where one frist initializes a smaller LM using the first few layers from larger pre-trained LM. Then, the smaller model is trained for a certain number of steps. Lastly, the smaller LM can be incrementally grown by adding more blocks of layer until pre-training loss is good or better than the large LM.

Stregnths:
- Clearly state problem (of attention collapse / lazy layer) and supoprt it with emerical results
- Propose solution to problem and run a series of experiments to show emperically models trained with Inheritune helps train models 1) good performance and 2) less attention collapse
- LLMs have been shown to be overparametarized. This work helps address this issue developing smaller models (which is nice for efficiency) that have good performance.

Weaknesses:
- Most evaluations are based only on the pre-training loss.
- Some minor typos

**Audience:**

Yes

**Audience Explanation:**

LLM's are known to be over-parametarized/redundant (i.e. it seems that a large portion of the parameters don't do much and mostly just help with training stability and giving the model more memorization power). I believe having methods for training smaller LLM's that converge faster with similar performance to larger model applicable to TMLR's audience.

**Broader Impact Concerns:**

Following https://jmlr.org/tmlr/ethics.html, I don't believe there are any concerns that would require a Broader Impact Statement.

**Claims And Evidence:**

Yes

**Claims Explanation:**

Author(s) provide clear and convicing evidence
- 1) that attention collapse occurs (Fig 1),
- 2) that attention collapse is indeed a problem (seciont 2.1, Fig 2)
- 3) that Inheritune helps mitigate this problem of attention collapse (Fig 8)
- 4) that Interitune helps trains smaller models with similar or better performances than training from scratch (large or same size model and warm started model) (Fig 6 and Table 2)

**Requested Changes:**

- Typos
    - Fig 7 caption: "Models derived using Inheritune without data repetition converges and matches"
        - converges -> converge
        - matches -> match
    - Page 12: "successful in training small LMs in some cases" -> "successful in training small LMs. In some cases"

- It would be nice if in Figure 8(a) and (b), the y-axis was on the same scale (i.e. both 0 - 40 or maybe 0 - 45). Or maybe just put them on the same graph with two different line colors/styles.
- Just for easy comparison, it would be nice to add in a "from scracth"/"random initialization" 16 layer model results at the beginning of table 3
- It would be nice to have some justification about mostly using the pre-training loss (i.e. does pre-training loss always correlate with downstream performance? Why only run downstream taks on one set of pre-training models and not pre-training models trained of OpenWeb)

---

> ### Author Response · Authors · 2025-12-25
> **Author's First Response to suggested revisions**
>
> Thank you for the positive assessment of our work. We tried our best to address all of your comments discussed below and also made changes in the paper draft (changes are marked in blue).
>
> ## Weakness
>
> > W1: Most evaluations are based only on the pre-training loss.
>
> **Based on your suggestion, we report an downstream evaluation of GPT-2 XLarge (1.5B) and variant(s) trained with OpenWebText (OWT) across nine tasks in Table 7.**
>
> We use pre-training loss as the primary evaluation metric because prior theoretical [1] and empirical [2] studies show correlation between training or validation loss with downstream in-context task performance. As a result, pre-training loss serves as a reliable indicator.
>
> Next, in this work, most GPT-2 models are trained on OpenWebText (OWT), following the original GPT-2 setup [3]. While OpenWebText includes Wikipedia articles, it lacks educational, scientific, and academic content, which is often crucial for strong performance on many downstream question-answering and common-sense tasks [4]. Additionally, we report downstream evaluations for models trained on FineWeb (FWB), which is widely recognized for its higher quality and extensive coverage of academic and educational documents, making it a more suitable target distribution for downstream evaluation.
>
> > W2: Some minor typos
>
> We have addressed the typos marked in blue in the current draft.
>
> ## Requested Changes
>
> * Typos fixed.
>
> * We have changed Figure 8 and Figure 12 and merged rank analysis of the full model and Ours (Our model developed with Inheritune) in one plot with different colors.
>
> * Please refer Table 5 in supplementary materials which already has all the val loss written which also include random init models.
>
> * We have answered this already in W1. Additionally, we have added Table 7, which thoroughly evaluates GPT-2 XLarge model and variant(s) trained with OWT on 9 different tasks.
>
> Thank you again for the thoughtful feedback.
>
> ## References
>
> [1] N. Saushi et al. ICLR 2020, A Mathematical Exploration of Why Language Models Help Solve Downstream Tasks.
>
> [2] M. Xia et al. ACL 2022, Training Trajectories of Language Models Across Scales.
>
> [3] A. Radford et al. 2019, Language Models are Unsupervised Multitask Learners.
>
> [4] F. Chen et al. 2025, The Coverage Principle: How Pre-Training Enables Post-Training.

---

> > ### Author Response · Authors · 2026-01-19
> > **We kindly request a response to our rebuttal**
> >
> > Dear Reviewer,
> >
> > We have addressed all of the changes you suggested and updated the manuscript accordingly. In particular, we have added additional downstream tasks in Table 7. We would appreciate it if you could let us know whether these revisions address your primary concerns.
> >
> > Thank you for your time and consideration.

---

> > > ### Comment · Reviewer_sKfL · 2026-01-19
> > > **Response to rebuttal**
> > >
> > > Yes and thank you! You have clearly addressed my concerns and I had already submitted the official recommendation based on all the revisions you made.

---

### Review · Reviewer_q1G7 · 2025-12-10

**Summary Of Contributions:**

The paper identifies attention collapse in decoder-only LLMs, showing that deeper layers often degenerate to near rank-1 attention maps, forming lazy layers that contribute little representational value (e.g., Figures 1–3).
To address this, the authors propose InheriTune, a progressive training recipe that initializes a compact model using early high-rank layers from a larger pretrained model and grows it until it matches the reference model's validation loss. Empirically, models trained through InheriTune achieve comparable or better performance than larger models and outperform both same-sized scratch-trained and warm-started baselines (Figures 5–7, Table 2).

Strengths:
* Clear empirical evidence of collapse
* Proposal of a simple and practical training method
* Carried out thorough baselines and analyzed useful ablations

Weaknesses:
* Limited theoretical explanation
* Restricted architectural diversity
* Potential computational overhead from multi-stage training

**Audience:**

Yes

**Audience Explanation:**

Understanding structural inefficiencies in transformers is highly relevant to researchers working on model compression and dilution, scaling laws, and efficient LLM training. The proposed method offers a practical alternative to pruning or distillation and is of broad interest.

**Broader Impact Concerns:**

Improving training efficiency for compact LLMs may accelerate widespread deployment, raising typical concerns around safety, misuse, and access to high-capability models. The Broader Impact section should briefly address risks from making strong models easier to train and distribute.

**Claims And Evidence:**

Yes

**Claims Explanation:**

Rank analyses, transfer experiments, and comparisons with strong baselines all support the central claims. Lazy layers are shown to behave like random initialization, confirming functional degeneration. Inheritune consistently matches or outperforms reference models across datasets and model sizes. The evidence is clear, though theoretical grounding could be deeper.

**Requested Changes:**

* Clarify and formalize the definition of lazy layers
Currently, layers are considered lazy when their attention matrices exhibit MaxRank(l) ≈ 1 (Figure 1) or AvgRank ≈ 1.2 (Figure 2).
Better Provide a precise threshold and analyze sensitivity to τ = 0.90 (used in rank calculation). Show how many layers become “lazy” under alternative τ values (e.g., 0.85, 0.95). This is important because conclusions about depth redundancy depend on the thresholding rule.

* Give deeper diagnostic or theoretical insight into why attention collapse arises
The manuscript references prior theory (Dong et al., Noci et al.), but observed collapse in GPT-2 occurs despite residuals, FFNs, and layernorms. Consider adding gradient-norm depth profiles for Q/K (as suggested in Section 2), or analyzing singular vector alignment over layers.
Data already shows full collapse in GPT-2 Medium's layers 20–24 (Figure 8a); explain why these layers fail to preserve rank.

* Quantify compute cost of Inheritune relative to baselines.
InheriTune requires multiple training rounds (e.g., GPT-2 Medium uses three rounds of 100K steps; Figure 19), whereas full baselines train once for 100K or 200K steps (Figures 5–6). Report total FLOPs or wall-clock to determine whether InheriTune truly improves overall efficiency, despite faster convergence of the final configuration.

* Expand evaluation to more modern architectures
The rank analysis on collapse is shown for GPT-2 families and limitedly for LLaMA-3 8B (Figure 10), but architectures with post-LN or parallel-attention (GPT-NeoX, MPT, Falcon) may behave differently. Adding results would strengthen claims of generality.
Include tasks where deeper layers are known to matter (e.g., BoolQ, Winogrande, NarrativeQA), to confirm that removing lazy layers does not degrade reasoning quality.

* Ablate the growth schedule in InheriTune
All experiments use +2 layers per round (Section 4). Test +1, +4, or adaptive increments to validate that results are not schedule-specific. Report how many rounds are required to match reference performance under each schedule.

---

> ### Author Response · Authors · 2025-12-25
> **Author's First Response to suggested revisions**
>
> Thank you for the positive and constructive feedback. We have addressed all comments below and incorporated the corresponding changes in the manuscript (marked in blue).
>
> ## Weakness
>
>
> >W1: Limited theoretical explanation
>
> **Mean-alignment (representation homogenization).**
>
> In deep Pre-LN transformers, residual connections preserve and repeatedly inject a shared representation across layers. After passing through multiple layers, token representations become increasingly similar (higher cosine similarity), i.e., a growing mean-aligned subspace dominates the per-token variation. In this regime, self-attention behaves like an iterative smoothing operator that performs near-uniform averaging: if token vectors are already highly aligned, then queries and keys become nearly identical across positions. Consequently, the attention score matrix QKᵀ / √d becomes approximately rank-1 (or low-rank), and softmax further amplifies this by producing nearly identical rows yielding near uniform averaging or a degenerate attention matrix (rank collapse).
>
> **Why deeper layers “get stuck” once collapse begins.**
>
> Once attention becomes close to uniform/degenerate, the layer’s output becomes less sensitive to individual token differences, which reduces the gradient signal needed to recover diverse, high-rank attention patterns. This creates a loop:
> token similarity → low-rank attention → weaker learning signal for W_Q, W_K → further collapse. This explanation aligns with prior theoretical results linking collapse to vanishing gradients of Q/K in deep layers, and it is consistent with our empirical finding that deeper layers in standard LLMs exhibit widespread near-rank-1 behavior.
>
> We will update the draft with a theoretical analysis in simplistic setting that discusses attention collapse and vanishing gradient phenomenon.
>
> >W2: Restricted architectural diversity.
>
> **To address this, we extend our attention rank analysis to four additional base LLMs: Falcon-7B, OLMo-1B, Cerebras GPT-2.7B, and LLaMA-3-3B.**
>
> Previously, our evaluation of attention collapse was largely limited to GPT-2 (Pre-LN) and LLaMA-3. We now broaden the analysis to cover MQA (with attention and MLP blocks in parallel) and the unusually large number of attention heads used in Falcon-7B. Cerebras GPT-2.7B and OLMo-1B employ MHA (with Cerebras GPT following Chinchilla scaling), while LLaMA-3-3B uses GQA and both Pre and Post RMS norm. See Section G.2 and Figures 20–21.
>
> >W3: Potential computational overhead from multi-stage training
>
> Indeed there could be some training overhead we have already acknowledged this in our limitations section. However if the bigger reference model has many lazy layers it will waste a lot of compute during inference and Inference-friendly models can be helpful in some specific use cases. Moreover, for GPT-2 Large and XLarge we have been able to achieve reduced depth variants with the same/better validation loss in just one round of Inheritune recipe.
>
> ## Requested Changes
>
> * To address this concern, we provided a more formal definition of lazy layers (rank based criterion) in Section G.6. Next, we also provide a sensitivity analysis in Section G.3 and Figure 22.
>
> * Refer our response to W1.
>
> * Compute Cost and Efficiency
>
>   * Large and XLarge (High Efficiency): For GPT-2 Large and XL, Inheritune matched or surpassed the reference loss in a single round of 100K steps. When compared to the same-sized baseline (200K steps trained from scratch), Inheritune represents a >50% reduction in total compute FLOPs, as it uses half the steps.
>   * Medium (Iterative Refinement): For the Medium variant (3 rounds), the total step count is higher. However, as shown in Table 5, the scratch baseline never reaches our loss floor even after 200K steps (2.83 vs. 2.81). Inheritune enables a smaller model to achieve a level of generalization (loss) that is effectively inaccessible to same-sized models trained from scratch.
>
> * We have added four new models (see Section G.2 and Figures 20–21). In addition, in Section G.1 and Table 7, we evaluate GPT-2 XL on nine downstream tasks, including three newly added tasks.
>
> * In Inheritune, at each round, we reinitialize the smaller (variant) model solely using the weights of the reference model. As a result, the method is agnostic to the growth schedule. For instance, adding four layers incrementally over two rounds or initializing all four layers at once leads to the same outcome. However, initializing more layers results in better generalization from the very beginning of training. Refer Table below.
>
> **Table. Validation loss of GPT-2 Large variants (100K steps using OWT) as a function of the number of initialized layers in Inheritune.**
>
> | Layer | Val Loss ↓ |
> |-------|------------|
> | l = 4 | 3.10       |
> | l = 8 | 2.89       |
> | l = 18| 2.80       |

---

> > ### Author Response · Authors · 2026-01-19
> > **We kindly request a response to our rebuttal**
> >
> > Dear Reviewer,
> >
> > Thank you for your thoughtful and constructive feedback. We have revised the manuscript to incorporate all of your suggestions. In particular:
> >
> > 1. We have added the requested theoretical analysis (Section H).
> > 2. We have included four additional models in the attention-rank analysis (Section G.2 and Figures 20–21).
> > 3. We have added a sensitivity analysis (Figure 22).
> >
> > We would be grateful if you could let us know whether these revisions have addressed your primary concerns.

---

### Review · Reviewer_Xk1j · 2025-12-10

**Summary Of Contributions:**

The authors identify that 'attention matrix rank collapse' occurs in the deeper layers of LLM architectures. They propose that these collapsed layers are functionally ineffective, and thus initialize shallower models using only the high-rank layers from a pre-trained model. They propose an iterative training algorithm to grow models based on these initial layers. They show that these shallower models can reach the performance of larger pretrained models with this recipe.


**Strengths:**
- The demonstration that models with partial-depth can converge to the same performance as pre-trained 'full-depth' models is interesting and novel to the best of my knowledge.
- The specific 'inheritune' training recipe which allows for this is also novel and is well described.
- The paper is written clearly and the experimental details appear complete.
- The inclusion of some ablations is important and helpful.

**Weaknesses:**
- The results of the 'initialization' experiments in Section 2.1 appear to have a potential confounding explanation which could invalidate the authors conclusion that 'lazy layers are functionally impaired'. Specifically, since 'lazy' layers are always deeper layers (as the authors have shown), it is very likely that simply transferring earlier layers is the key for strong initialization initialization, and the 'lazy' low-rank property could be irrelevant. This would also seem to be backed up by the red line in Figure 2b surpassing the yellow line which has higher rank.
	- In my opinion, one way to partially alleviate this confounder would be to perform the initialization experiments of only deeper layers in a deeper network, leaving the early layers random. However this is still only a partial signal for the functional relevance of the layers and I believe fundamentally something beyond initialization experiments should be done to make this claim.
- The experimental results in Figures 5 & 6 are a bit misleading since the 'training time' requires not only the training time of full model, but also the training times of the iterative refinement steps. For example, are the 2x speedups reported in Figure 6 completely overshadowed by the fact that the models need to be iteratively retrained more than 2 times?
- There are no error bars on any plots or tables. This is worrisome for results such as those reported in Table 2 where the performance gain is marginal.
- Many of the models do not appear fully converged in training (e.g. Fig 2b, 5). The choice of 100k training steps for the base model appears somewhat arbitrary.
- The comparison of 'inheritune' with the more common alternative 'hybrid-stacking' in Table 2 is challenging. Specifically, the results with the X-large model bring into question the benefit the iterative procedure of inheritune, and if it outweighs the multiplicative increase in training time.
- The authors appear to ignore the fact that their pre-trained initialization is a huge reason for their faster convergence, rather than some deeper theoretical reason. For example, comments such as the following should be removed:
	- "From a convergence perspective, prior work has linked overparameterization to faster convergence Bengio et al. (2005); Vaswani et al. (2018). Interestingly, we find that smaller models derived using Inheritune converge just as fast as their larger counterparts."
- The results are largely overstated in the discussion section. For example, the authors state: "While stacking and hybrid stacking demonstrate reasonable performance, they still fall short compared to Inheritune. Across all cases, Inheritune consistently outperforms these baselines, highlighting its effectiveness as an initialization strategy." Yet, in Table 2, they show that hybrid-stacking performs equivalently to inheritune for the XL mode(2.64 vs. 2.64) -- this is not 'outperforming'.
- Figure 8 does not appear to support the authors claim that 'inheritune mitigates attention collapse'. The difference in x-axis scale seems to account for the majority of difference in the plots -- i.e. if you only look at the first 16 layers of Fig 8a, it appears to match Fig 8b. If there is truly a difference, the authors should overlay the plots and quantify this difference precisely.
- I believe there is a relevant citation that should be included, which also discusses attention rank collapse is causal transformers.
	- Wu et al., “On the Role of Attention Masks and LayerNorm in Transformers” (NeurIPS 2024)

**Additional Comments:**

**Minor:**
- Figure 3b says (16-layer) but the caption says (18-layer)
- The description of the inheritune algorithm is a bit odd. While it makes sense, the authors should avoid using ambiguous phrases such as: "repeat steps 1-2 until desired performance is achieved." Clearly the desired performance is as high as possible, so it would be more appropriate to say something like 'repeat until convergence or a decrease in performance'.
- The writing in the paper overall is a bit informal and would benefit from both being less verbose and more precise.

**Audience:**

Yes

**Audience Explanation:**

The result that you can achieve the performance of a deeper network with fewer layers is likely of interest to some readers, regardless of training time. The proposed Inheritune algorithm indeed does appear to surpass the performance of comparable alternative (e.g. hybrid stacking) for smaller models (but not for larger models). The authors would do better to emphasize this benefit and not over state their results or conclusions.

**Claims And Evidence:**

No

**Claims Explanation:**

As outlined in the weaknesses listed above, there are multiple claims that are only partially supported by evidence, and there may be confounding factors that could invalidate the conclusions. For example, the claims 'lazy layers are functionally ineffective' and 'inheritune mitigates attention collapse', are not fully supported by the experiments the authors perform. Furthermore, the claim that inheritune yields a 2x speedup in training is largely misleading (as the authors note very subtly in the limitations section) since they must retrain the model multiple times prior to achieving this speedup.

**Requested Changes:**

While the phenomenon of attention collapse is important, the experimental evaluation of the 'inheritune' method has a number of potential confounders and conflicting results with respect to the authors claimed conclusions. I therefore recommend the authors carefully revise their claims (as listed in the weaknesses), or improve their experimental evaluation to increase the evidence for their existing conclusions. These changes would be required for the paper to be ready for acceptance in my view.



**Questions:**
- Do the authors disagree with the first noted weakness that layer depth may be a potential confounding factor?
- Do the authors understand why in Figure 2b, the 'init w/ middle four layers' appears to work better than the 'init w/ first four layers', despite these layers appearing more 'lazy'? This would seem to contradict the authors' main point with this figure.
 - Can the authors justify the choice of 100k training steps for the base model?
- Can the authors comment on the relation of their findings to the Wu et al. 2024 paper?

---

> ### Author Response · Authors · 2025-12-26
> **Author's First Response to comments and suggested revisions (Part 1)**
>
> We thank the reviewer for their thoughtful and detailed feedback. We are encouraged that you found our demonstration of partial-depth models matching full-depth performance to be “interesting and novel” and our training recipe “well described.” All changes in the revised draft are highlighted in blue.
>
> ## Weakness
>
> 1. **Confounding Factor: Depth vs. Rank**
>
> The reviewer suggests that the success of initialization might be due to layer “earliness” rather than rank. However, our results in Figure 2b and Table 4 actually provide evidence for the rank hypothesis over the depth hypothesis:
>
> * Rank over Proximity: In Figure 2b, the red line (Middle layers 5-8, AvgRank 9.48) outperforms the yellow line (First layers 1-4, AvgRank 8.40). If  the early layers (proximity to the embedding layer) were the primary driver, the first four layers should have performed best. Instead, the layers with the highest Avgrank (middle layers) yielded the best initialization, despite being further from the embedding.
>
> * Lazy vs. Random: The “lazy” layers (layers 9-12, AvgRank 1.22) perform quite close to random initialization. This suggests that once a layer collapses to rank-1, it loses its “pre-trained advantage,” regardless of its position. We have explicitly stated that our evidence supports poor transferability of rank-collapsed layers, and not rank as the sole causal factor behind a layer’s utility.
>
> We have defined the “functional ineffectiveness” in a limited scope, i.e. if a layer’s contribution can be entirely replaced by a smaller, higher-rank stack without losing generalization, and if that layer performs as poorly as random weights when transferred, it is functionally impaired relative to the rest of the model.
>
> * Addressing the requested experiment
>
> **Please refer Section G.4 and Figure 23.** The result provides further evidence that that initializing with lazy layers consistently degrades performance.
>
> We agree that a deeper mechanistic analysis of layer importance using rank would be highly interesting; however, it is beyond the scope of our current work and we defer it to future.
>
> 2. **Training Time and Iterative Refinement (Figures 5 & 6)**
>
> One-Round Parity: For GPT-2 Large (770M) and XL (1.5B), we achieved parity in a single round without growth phase. For these models, the 2x speedup is literal. Only the Medium variant required three rounds, which we explicitly note as limitation.
>
> Even for GPT-2 Medium, which requires three rounds, the total training budget (100k + 100k + 100k steps) yields a model that matches the performance of a 24-layer reference. In contrast, training a same-sized 16-layer model from scratch for 200k steps still exhibits a clear gap (Val. Loss 2.83 vs. 2.81 for Inheritune; Table 5). Thus, the iterative cost is an investment that achieves a lower loss floor that scratch training struggles to reach.
>
> For models that require multiple rounds (like GPT-2 Medium), Inheritune is not just a faster way to train; it is a way to achieve a better generalization floor. As shown in Table 5 (Figure 6), a 16-layer model trained from scratch for 200K steps (twice the budget) still exhibits a clear performance gap compared to the Inheritune version (Val. Loss 2.83 vs. 2.81).
>
> Training Gains in extended regime: In Figure 24, we demonstrate that an 18-layer variant trained via Inheritune for 200K steps actually outperforms the 24-layer full model baseline (Val. Loss 2.86 vs 2.87). This reinforces that the "iterative cost" is an investment in model quality that scratch training fails to match.
>
> 3. **Statistical Rigor and Convergence**
>
> We have followed established experimental practice for GPT-2–scale models ([1, 2,3]). Typically, in LLM training literature error bars for training curevs are not shown as this is just single round of training and it is quite expensive to train multiple times. Similarly for downstream tasks error bars are not provided (refer GPT-2 paper [1]). To address some of these concerns **we have added std for the downstream tasks (Table 7)** to demonstrate that Inheritune’s performance gains are robust.
>
> In Table 2 we report final log loss the gains numbers (refer full training plots Figure 17) are not marginal.  For instance, a log-loss reduction from 2.89 to 2.80 (Table 2, GPT-2 Large) represents a ~10% improvement in Perplexity (18.0 → 16.4). In LLM setting unlike ViTs we don’t train with many epochs (i.e. with limited repetition) so, convergence through saturation is not possible. The 100K steps were used to utilize the full 9B tokens of the train set (amd beyond). **Added a new result with extended training regime with 200K train steps (refer Figure 24).**
>
> (Contd ...)

---

> ### Author Response · Authors · 2025-12-26
> **Author's First Response to comments and suggested revisions (Part 2)**
>
> 4. **Comparison with Hybrid-Stacking (Table 2 ) and overstatement**
>
> Both the stacking [4] and the hybrid stacking method uses iterative refinement by growing the model during training (layerwise), so both Inheritune and hybrid stacking may have multiplicative increase in training time. Hybrid stacking is a method we created for apples to apples comparison.
>
> * Overall, in Table 2 Inheritune performs better: GPT-2 XL Parity: For the 1.5B XL model, Inheritune achieved parity with Hybrid-Stacking (2.64). However, for GPT-2 Medium and Large, Inheritune consistently achieved the lowest validation loss (Medium: 2.81 vs 2.83; Large: 2.80 vs 2.89). We have revised the draft to acknowledge this.
> * We agree that pre-trained initialization is the primary driver for fast convergence. However, the insight is that not all pre-trained layers are equal. Picking the “wrong” layers (lazy layers) results in performance no better than random initialization (Figure 3). We have revised the text accordingly.
>
>
> 5. **Inheritune's contribution towards mitigating attention collapse**
>
> Thank you for the suggestion. We have combined the two figures into a single consolidated figure (see Figures 8 and 12).
>
> We would also like to clarify the key observation highlighted in Figure 8. The full reference model exhibits a substantial number of collapsed (lazy) layers beyond the halfway depth. In contrast, the Inheritune model → initialized by inheriting the first 16 layers from the reference model and then trained for 100K steps shows no lazy layers beyond the halfway mark. This directly illustrates inheritune helps in mitigating attention collapse.
>
> Thank you for pointing out the work in [6], which is indeed highly relevant, and we will include a citation. That paper’s theoretical analysis studies rank dynamics while omitting residual connections and MLP blocks, and it explores a complementary direction using sparse masking with LayerNorm to prevent rank collapse. Our work, by contrast, focuses on a practical, empirical setting: we demonstrate that rank collapse persists in modern LLMs and show that initialization via inheritance and stage-wise training provides an effective mechanism to mitigate this phenomenon in practice.
>
> ## Requested Changes
>
> We have responded to the weaknesses. We have extended our attention rank analysis to four additional open weight base LLMs: Falcon-7B, OLMo-1B, Cerebras GPT-2.7B, and LLaMA-3-3B (refer to Section G.2). We have also added further evidences to our claims.
>
> ## Questions
>
> * Yes we disagree and have provided clarification in W1.
>
> * This statement is wrong. The middle layers have higher AvgRank (9.48) compared to early layers (8.40) and none of them are lazy. It aligns with our hypothesis.
>
> * Refer W3.
>
> * Refer W5.
>
> ## Additional Comments
>
> Fixed the typos and made some changes.
>
> ## References
>
> [1] Language Models are Unsupervised Multitask Learners.
>
> [2] Sophia: A Scalable Stochastic Second-order Optimizer for Language Model Pre-training.
>
> [3] Early Weight Averaging meets High Learning Rates for LLM Pre-training.
>
> [4] Efficient Training of BERT by Progressively Stacking.
>
> [5] LLaMA: Open and Efficient Foundation Language Models.
>
> [6] On the Role of Attention Masks and LayerNorm in Transformers.

---

> ### Comment · Reviewer_Xk1j · 2026-01-08
>
> I thank the authors for their thorough response.
>
> I appreciate that they have corrected my misunderstanding with respect to Figure 2b and the potential confounder; they have alleviated my concern in this respect.
>
> I additionally appreciate their refined claims and updates to the text.
>
> Figure 23 is a welcome addition which indeed partially helps to support their claims that lazy layers are ineffective. While I think that more should be done to fully validate this claim, they have provided sufficient evidence for this limited 'initialization' experimental setting.
>
> Figure 24 is also appreciated, showing that longer training still yields gains, but the performance benefits are definitely reduced compared to what is shown in the main text. I believe this result should be mentioned more prominently to not mislead readers.
>
> The authors note: "For GPT-2 Large (770M) and XL (1.5B), we achieved parity in a single round without growth phase. For these models, the 2x speedup is literal." I still think this is still an overstatement -- the full size model must be trained first in order to perform the initialization. This again should be stated very clearly in the text to not mislead readers.
>
> The new Figure 8 is significantly improved and much appreciated, however again the 'mitigation of rank collapse' is only very weakly supported by this figure. The differences in layer rank are marginal and potentially within random fluctuations.
>
> Overall, I thank them for taking my review seriously and providing the additional results.

---

> > ### Author Response · Authors · 2026-01-11
> > **Author's Second Response**
> >
> > We thank the reviewer for the helpful and constructive feedback.
> >
> >
> >
> > > I appreciate that they have corrected my misunderstanding with respect to Figure 2b and the potential confounder; they have alleviated my concern in this respect. I additionally appreciate their refined claims and updates to the text.
> >
> > > Figure 23 is a welcome addition which indeed partially helps to support their claims that lazy layers are ineffective. While I think that more should be done to fully validate this claim, they have provided sufficient evidence for this limited 'initialization' experimental setting.
> >
> > We are glad that we have addressed these concerns.
> >
> >
> > >  Figure 24 is also appreciated, showing that longer training still yields gains, but the performance benefits are definitely reduced compared ...
> >
> > Thank you for the suggestion. We will explicitly highlight this point. While Figure 24 shows that performance gains from longer training are indeed reduced relative to those reported in the main text, models trained with Inheritune still perform on par with the corresponding full model. This observation continues to support our central claim, even though the full model exhibits slightly lower overall performance under extended training.
> >
> >
> >
> > > The authors note: "For GPT-2 Large (770M) and XL (1.5B), we achieved parity in a single round without growth phase. For these models, the 2x speedup is literal." ...
> >
> >
> >
> > We will explicitly add this as a limitation in the revised manuscript. We re-emphasize two points. First, there is a clear trade-off: models such as GPT-2 Medium achieve a lower loss floor than same-sized models trained for 100K–200K steps from scratch. Second, in practice, organizations often train multiple model sizes independently (for example, the Gemma family with 27B, 12B, 4B, 1B, and 270M variants, all trained from random initialization). In such settings, Inheritune can be directly applied and remains practically useful.
> >
> >
> >
> > >  The new Figure 8 is significantly improved and much appreciated, however again the 'mitigation of rank collapse' is only very weakly supported by this figure. The differences in layer rank are marginal and potentially within random fluctuations.
> >
> >
> > We respectfully disagree. In Figure 8, we analyze rank collapse in deeper layers beyond the halfway point. For the full 24-layer model, 3 out of 12 deeper layers are lazy, with layers beyond 15 visually collapsing close to maximum rank-1. In contrast, none of the deeper layers in our 16-layer Inheritune model exhibit lazy behavior.
> >
> > This trend is even more pronounced in Figure 12. For the 48-layer full model, 22 deeper layers beyond the halfway point are lazy, whereas the Inheritune-trained model exhibits only 2 such layers (out of 12). These results indicate a clear and consistent mitigation of rank collapse.

---

### Decision · Action_Editor_X9wB · 2026-01-25

**Recommendation:** Accept as is

**Audience:**

Yes

**Audience Explanation:**

This paper studies efficient LLM training, this is at the core of ML research these days, hence I expect this paper to be relevant to a big portion of the community.

**Claims And Evidence:**

Yes

**Claims Explanation:**

This paper studies attention initialization in Transformer-based LLMs and identifies rank collapse in deeper layers, which the authors argue are largely ineffective. They propose initializing shallower models using high-rank layers from pretrained models and introduce an iterative training procedure to grow such models. Experiments show that this approach allows smaller models to match the performance of larger pretrained ones. The claims are well supported by empirical evidence and prior work, and the authors adequately addressed reviewer concerns. I recommend acceptance.